# 3BASiL: An Algorithmic Framework for Sparse plus Low-Rank Compression of LLMs

**Mehdi Makni**
Operations Research Center
Massachusetts Institute of Technology
mmakni@mit.edu

**Xiang Meng**
Operations Research Center
Massachusetts Institute of Technology
mengx@mit.edu

**Rahul Mazumder**
Operations Research Center
Massachusetts Institute of Technology
rahulmaz@mit.edu

## Abstract

Sparse plus Low-Rank ($\mathbf{S} + \mathbf{LR}$) decomposition of Large Language Models (LLMs) has emerged as a promising direction in *model compression*, aiming to decompose pre-trained model weights into a sum of sparse and low-rank matrices $\mathbf{W} \approx \mathbf{S} + \mathbf{LR}$. Despite recent progress, existing methods often suffer from substantial performance degradation compared to dense models. In this work, we introduce `3BASiL-TM`, an efficient one-shot post-training method for $(\mathbf{S} + \mathbf{LR})$ decomposition of LLMs that addresses this gap. Our approach first introduces a novel 3-Block Alternating Direction Method of Multipliers (ADMM) method, termed `3BASiL`, to minimize the layer-wise reconstruction error with convergence guarantees. We then design an efficient transformer-matching (`TM`) refinement step that jointly optimizes the sparse and low-rank components across transformer layers. This step minimizes a novel memory-efficient loss that aligns outputs at the transformer level. Notably, the `TM` procedure is universal as it can enhance any $(\mathbf{S} + \mathbf{LR})$ decomposition, including pure sparsity. Our numerical experiments show that `3BASiL-TM` reduces the WikiText2 perplexity gap relative to dense LLaMA-8B model by over $30\%$ under a (2:4 Sparse + 64 LR) configuration, compared to prior methods. Moreover, our method achieves over 2.5x faster compression runtime on an A100 GPU compared to SOTA $(\mathbf{S} + \mathbf{LR})$ method. Our code is available at https://github.com/mazumder-lab/3BASiL.

## 1 Introduction

Large Language Models (LLMs) have demonstrated exceptional performance across diverse tasks including complex reasoning [Xu et al., 2025], text generation [Achiam et al., 2023], mathematical problem-solving [Romera-Paredes et al., 2024], and code synthesis [Roziere et al., 2023]. However, state-of-the-art LLMs [Achiam et al., 2023, Dubey et al., 2024, Google, 2023] with billions of parameters face substantial deployment challenges due to their computational and memory requirements. These constraints substantially limit real-time applications and deployment on resource-constrained devices. Consequently, model compression techniques have emerged as an essential research direction to increase LLM accessibility while preserving their accuracy and functionality.

Established methods for *model compression* primarily include neural network pruning [LeCun et al., 1989, Hassibi and Stork, 1992, Han et al., 2015b] and quantization [Han et al., 2015a, 2016]. For LLMs, recent research has focused on one-shot post-training compression methods [Frantar and

39th Conference on Neural Information Processing Systems (NeurIPS 2025).

Alistarh, 2023, Dettmers et al., 2023, Lin et al., 2024, Frantar et al., 2022, Behdin et al., 2023, Meng et al., 2024a,b] that compress model weights using a minimal calibration dataset without expensive retraining. These approaches have become particularly attractive as they enable efficient compression of modern LLMs even on a single commodity GPU.

An exciting recent line of research in one-shot compression studies the task of decomposing pre-trained weight matrices $\mathbf{W}$ into a compressed backbone component (e.g., sparse or quantized) and a low-rank component: $\mathbf{W} \approx \mathcal{C}(\mathbf{W}) + \mathbf{LR}$. This *LoRA-aware* formulation effectively integrates with Low-Rank Adaptation (LoRA) methods [Hu et al., 2022], allowing efficient downstream adaptation by freezing $\mathcal{C}(\mathbf{W})$ and fine-tuning only the low-rank components, which serve as a *smart-initialization* to LoRA. Guo et al. [2024], Li et al. [2024] demonstrate that this approach outperforms the *sequential* approach of first compressing the model $\mathbf{W} \approx \mathcal{C}(\mathbf{W})$ (under similar backbone components constraints) followed by standard LoRA fine-tuning (*Noise & Zero* adapter).

One-shot Quantized plus Low-Rank decomposition methods for LLMs [Guo et al., 2024, Li et al., 2024, Saha et al., 2024] have demonstrated exceptional efficiency. These works decompose pre-trained LLM matrices into low-rank components and memory-efficient quantized backbones, enabling aggressive quantization while preserving model performance.

Parallel advances in Sparse plus Low-Rank $(\mathbf{S} + \mathbf{LR})$ decomposition [Zhang and Papyan, 2025, Makni et al., 2025] combine the strengths of pruning and matrix factorization. OATS [Zhang and Papyan, 2025] pioneered post-training $(\mathbf{S} + \mathbf{LR})$ compression for LLMs, demonstrating its viability as an alternative to unstructured pruning at the same compression rates and achieving CPU acceleration via DeepSparse [NeuralMagic, 2021]. HASSLE-Free [Makni et al., 2025] established a unified $(\mathbf{S} + \mathbf{LR})$ framework showing an underlying connection between OATS and various LLM pruning methods—they focus on N:M sparse plus low-rank decompositions for GPU acceleration using specialized CUDA kernels from [Mozaffari et al., 2024]. Despite their promise, existing $(\mathbf{S} + \mathbf{LR})$ algorithms for LLMs rely exclusively on alternating minimization approaches. Due to the complexity of the underlying optimization problem, these procedures have limited convergence guarantees and may perform poorly in *joint* optimization of the sparse and low-rank components. Indeed, our empirical evidence in Figure 5a and Figure 5b suggests that our proposed algorithm more effectively optimizes the sparse and low-rank parts compared to an alternating minimization approach.

In this paper, we propose `3BASiL`[1], an elegant 3-block ADMM approach tailored for $(\mathbf{S} + \mathbf{LR})$ decomposition of LLMs. Unlike prior approaches that separate pruning and low-rank fitting steps, `3BASiL` explicitly models their interaction through simultaneous optimization under a unified objective with provable convergence guarantees. We formulate the weight decomposition problem with explicit sparsity and rank constraints, decomposing it into three variable sets—sparse component, low-rank component, and original weights—optimized within an iterative ADMM framework. This approach precisely enforces sparsity pattern and rank constraints at each iteration via closed-form proximal updates while minimizing reconstruction error with respect to the original model weights.

Additionally, we propose a (memory-efficient) transformer-matching (`TM`) procedure that refines sparse and low-rank components by aligning transformer block outputs with the dense model. In contrast to prior $(\mathbf{S} + \mathbf{LR})$ methods, which allow only low-rank components refinement via LoRA after layerwise compression, `TM` enables joint refinement of both sparse and low-rank components at the transformer level. This procedure is compatible with any existing $(\mathbf{S} + \mathbf{LR})$ method and can be applied prior to LoRA fine-tuning with minimal computational overhead, providing a more effective initialization for downstream adaptation.

Our **contributions** can be summarized as follows:

1. **3-Block ADMM** We introduce `3BASiL`, a novel 3-Block Alternating Direction Method of Multipliers (ADMM) algorithm specifically designed for Sparse plus Low-Rank $(\mathbf{S} + \mathbf{LR})$ decomposition of Language Models. Our method explicitly captures interactions between sparse and low-rank components within a unified optimization framework, while providing theoretical convergence guarantees as well. Moreover, `3BASiL` offers remarkable computational advantages, achieving over 7x speedup compared to the strong HASSLE-free-ALPS baseline, when compressing a Llama3.2-3B model on an L40 48GB GPU.

---

[1]`3BASiL`: **3**Block **A**DMM for **S**parsity and **L**ow-Rank Constraints.

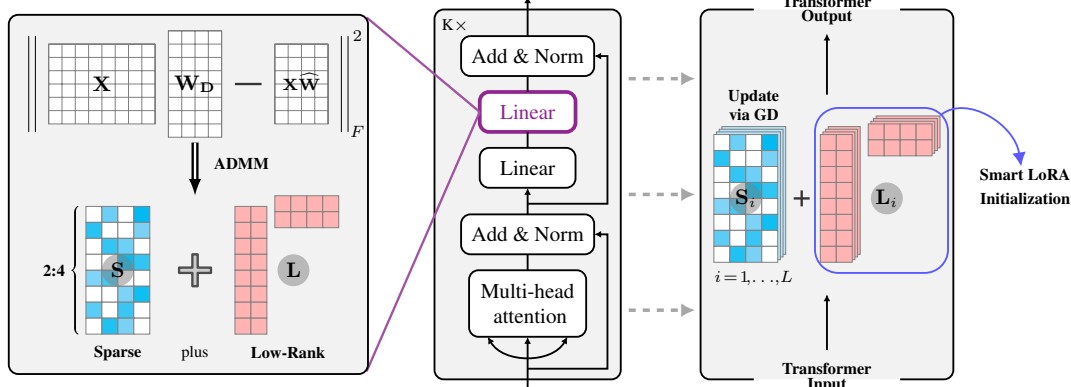

Figure 1: Overview of the proposed 3BASiL framework. (**Left**) For each layer in a Transformer, we employ multi-Block ADMM to efficiently decompose weights into high-quality Sparse plus Low-Rank components by minimizing the layer reconstruction objective. (**Right**) At the Transformer level, we apply gradient-based optimization to jointly refine all sparse and low-rank components across layers to match the original transformer's output, with the resulting low-rank components serving as smart initialization for subsequent LoRA fine-tuning.

2. **Transformer matching and Universality**  We introduce TM, a novel (memory-efficient) refinement procedure that jointly optimizes sparse and low-rank components across transformer layers. This approach significantly improves sparse component quality with minimal computational cost by directly leveraging transformer-level outputs, addressing a major limitation in current $(\mathbf{S} + \mathbf{LR})$ methods. Crucially, our TM procedure is universally applicable and can enhance any existing $(\mathbf{S} + \mathbf{LR})$ decomposition method, including purely sparse compression, providing superior initialization for subsequent LoRA fine-tuning.

3. **Empirical Validation and State-of-the-Art Results**  We introduce 3BASiL-TM as a new state-of-the-art method for $(\mathbf{S} + \mathbf{LR})$ one-shot decomposition of Large Language Models. It significantly improves LLM evaluation benchmarks including perplexity of different datasets and various zero-shot tasks. Specifically, our numerical experiments show that 3BASiL-TM reduces the WikiText2 perplexity gap to dense model by over $30\%$ compared to prior methods for a Llama-8B model under a (2:4 Sparse + 64 LR) configuration. It also provides significant compression runtime speedups compared to other $(\mathbf{S} + \mathbf{LR})$ decomposition techniques for LLMs.

## 2 Highly effective Sparse plus Low-Rank decomposition via ADMM

### 2.1 Problem formulation

We compress the layers of an LLM sequentially, one at a time by minimizing the reconstruction error between the outputs of pre-trained weights and compressed ones on a set of given input activations. Formally, let $\widehat{\mathbf{W}}$ represent the pre-trained weight matrix of a given layer, and $\mathbf{X}$ denote the input activations (i.e., output of previous layers) on a set of $N$ calibration samples. In our setting, the goal of layer-wise reconstruction is to find a $(\mathbf{S} + \mathbf{LR})$ decomposition that minimizes the $\ell_2$ error between the outputs of the original and decomposed weights—this can be formulated as follows:

$$\min_{\mathbf{S},\mathbf{L}} \quad \frac{1}{2}\left\|\mathbf{X}\widehat{\mathbf{W}} - \mathbf{X}\left(\mathbf{S} + \mathbf{L}\right)\right\|_F^2 + \frac{\lambda}{2}\left\|\widehat{\mathbf{W}} - \left(\mathbf{S} + \mathbf{L}\right)\right\|_F^2 \quad \text{s.t.} \quad \mathbf{S} \in \mathcal{S}, \quad \text{rank}\left(\mathbf{L}\right) \leq r. \quad (1)$$

Above $\|\cdot\|_F$ denotes the Frobenius norm, $\mathcal{S}$ denotes the set of matrices satisfying a specified sparsity constraint (e.g., unstructured sparsity with given sparsity level or N:M sparsity); $\mathbf{S}$ and $\mathbf{L}$ denote the sparse and low-rank components, respectively. Parameter $\lambda > 0$ encourages the decomposed weights to remain close to the pre-trained ones.

### 2.2 A multi-Block ADMM approach for layer-wise reconstruction

The primary challenge in optimizing problem (1) lies in the joint minimization of $\mathbf{S}$ and $\mathbf{L}$ under two complex constraints—sparsity and low-rank. To address this, we employ the Alternating Direction Method of Multipliers (ADMM), which enables separate updates of $\mathbf{S}$ and $\mathbf{L}$ at each iteration while

maintaining their interdependence through a Lagrangian multiplier. This approach preserves the power of joint optimization while making the problem tractable. Our 3-block ADMM introduces an auxiliary variable $\mathbf{D}$ as a copy of the sparse component $\mathbf{S}$, reformulating problem (1) as:

$$\min_{\mathbf{S},\mathbf{D},\mathbf{L}} \quad \frac{1}{2}\left\|\mathbf{X}\widehat{\mathbf{W}} - \mathbf{X}(\mathbf{S}+\mathbf{L})\right\|_F^2 + \frac{\lambda}{2}\left\|\widehat{\mathbf{W}} - (\mathbf{S}+\mathbf{L})\right\|_F^2 + \mathbb{I}_{\mathcal{S}}(\mathbf{D}) \tag{2}$$
$$\text{s.t.} \quad \mathbf{S} = \mathbf{D}, \quad \text{rank}(\mathbf{L}) \leq r.$$

where $\mathbb{I}_{\mathcal{S}}(\mathbf{D})$ is an indicator function that equals to infinity when $\mathbf{D} \notin \mathcal{S}$ and zero otherwise. The augmented Lagrangian function with dual variable $\mathbf{V}$ and a quadratic penalty parameter $\rho > 0$ reads:

$$\mathcal{L}_\rho(\mathbf{S},\mathbf{L},\mathbf{D},\mathbf{V}) = \frac{1}{2}\left\|\mathbf{X}\widehat{\mathbf{W}} - \mathbf{X}(\mathbf{S}+\mathbf{L})\right\|_F^2 + \frac{\lambda}{2}\left\|\widehat{\mathbf{W}} - (\mathbf{S}+\mathbf{L})\right\|_F^2 + \mathbb{I}_{\mathcal{S}}(\mathbf{D}) + \frac{\rho}{2}\left\|\mathbf{S}-\mathbf{D}+\frac{\mathbf{V}}{\rho}\right\|_F^2.$$

The method proceeds by minimizing the augmented Lagrangian with respect to three variables sequentially: the sparse component $\mathbf{S}$, the low-rank component $\mathbf{L}$, and sparse component's constrained copy $\mathbf{D}$, followed by a dual update (in variable $\mathbf{V}$). This sequential optimization over three variable blocks gives the method its name: 3-Block ADMM. At iteration $t$, the updates are:

$$\mathbf{S}^{(t+1)} = \arg\min_{\mathbf{S}} \mathcal{L}_\rho(\mathbf{S},\mathbf{L}^{(t)},\mathbf{D}^{(t)},\mathbf{V}^{(t)}) \qquad \mathbf{L}^{(t+1)} = \arg\min_{\mathbf{L}} \mathcal{L}_\rho(\mathbf{S}^{(t+1)},\mathbf{L},\mathbf{D}^{(t)},\mathbf{V}^{(t)})$$
$$\mathbf{D}^{(t+1)} = \arg\min_{\mathbf{D}} \mathcal{L}_\rho(\mathbf{S}^{(t+1)},\mathbf{L}^{(t+1)},\mathbf{D},\mathbf{V}^{(t)}) \quad \mathbf{V}^{(t+1)} = \mathbf{V}^{(t)} + \rho(\mathbf{S}^{(t+1)} - \mathbf{D}^{(t+1)}).$$

Below, we derive the updates. For notational simplicity, we denote $\mathbf{H} = \mathbf{X}^\top\mathbf{X} + \lambda\mathbf{I}$.

**S-block update** Since $\mathcal{L}_\rho(\mathbf{S},\mathbf{L}^{(t)},\mathbf{D}^{(t)},\mathbf{V}^{(t)})$ is a quadratic function of $\mathbf{S}$, we obtain the closed-form solution by setting the gradient to zero:

$$\mathbf{S}^{(t+1)} = (\mathbf{H}+\rho\mathbf{I})^{-1}\left(\mathbf{H}(\widehat{\mathbf{W}}-\mathbf{L}^{(t)}) - \mathbf{V}^{(t)} + \rho\mathbf{D}^{(t)}\right). \tag{3}$$

**L-block update** Note that the $\mathbf{L}$-optimization subproblem can be reformulated as minimizing $\|\mathbf{H}^{1/2}(\widehat{\mathbf{W}}-\mathbf{S}^{(t+1)}-\mathbf{L})\|_F^2$ subject to the rank constraint. When $\mathbf{H}$ is full-rank (satisfied for any $\lambda > 0$), this problem has the closed-form solution (see Section 5 for a discussion about rank-reduced regression results) :

$$\mathbf{L}^{(t+1)} = \mathbf{H}^{-1/2}P_r\left(\mathbf{H}^{1/2}(\widehat{\mathbf{W}}-\mathbf{S}^{(t+1)})\right), \tag{4}$$

where $P_r$ denotes the best rank-$r$ approximation, which can be computed via SVD.[2]

**D-block update** The optimization over $\mathbf{D}$ involves projecting $\mathbf{S}^{(t+1)} + \mathbf{V}^{(t)}/\rho$ onto the sparsity constraint set $\mathcal{S}$, which corresponds to magnitude-based pruning of $(\mathbf{S}^{(t+1)} + \mathbf{V}^{(t)}/\rho)$—we sort $[(\mathbf{S}^{(t+1)} + \mathbf{V}^{(t)}/\rho)_{ij}]^2$ and retain only the largest values. For unstructured pruning, we keep a predetermined fraction of the largest values; for N:M structured sparsity, we retain N largest values out of every M consecutive weights.

In practice, we employ an iteration-dependent penalty parameter $\rho_t$, giving the following updates:

$$\mathbf{S}^{(t+1)} = (\mathbf{H}+\rho_t\mathbf{I})^{-1}(\mathbf{H}(\widehat{\mathbf{W}}-\mathbf{L}^{(t)}) - \mathbf{V}^{(t)} + \rho_t\mathbf{D}^{(t)}) \quad \mathbf{L}^{(t+1)} = \mathbf{H}^{-1/2}P_r(\mathbf{H}^{1/2}(\widehat{\mathbf{W}}-\mathbf{S}^{(t+1)}))$$
$$\mathbf{D}^{(t+1)} = P_{\mathcal{S}}(\mathbf{S}^{(t+1)} + \mathbf{V}^{(t)}/\rho_t) \qquad\qquad\qquad \mathbf{V}^{(t+1)} = \mathbf{V}^{(t)} + \rho_t(\mathbf{S}^{(t+1)} - \mathbf{D}^{(t+1)}). \tag{5}$$

**Computational complexity** We implement several tricks to reduce the computational cost in the $\mathbf{S}$ and $\mathbf{L}$-update steps, which constitute the major computational cost of 3-Block ADMM algorithm. For the $\mathbf{S}$-update step, we adopt the approach of Meng et al. [2024a] by pre-computing (once) and storing the eigenvalue decomposition $\mathbf{H} = \mathbf{U}\mathbf{\Sigma}\mathbf{U}^\top$. This allows us to efficiently calculate the matrix inverse $(\mathbf{H}+\rho\mathbf{I})^{-1} = \mathbf{U}(\mathbf{\Sigma}+\rho\mathbf{I})^{-1}\mathbf{U}^\top$ for varying values of $\rho$ across iterations. For an efficient $\mathbf{L}$-update step, we store the matrices $\mathbf{H}^{-1/2} = \mathbf{U}\mathbf{\Sigma}^{-1/2}$, and $\mathbf{H}^{1/2} = \mathbf{\Sigma}^{1/2}\mathbf{U}^\top$ and employ a randomized-SVD procedure [Halko et al., 2011] for numerical efficiency. In the context of LLMs, the weight matrices scale with the transformer's hidden dimension $N$. Our algorithm's per-iteration time complexity comprises: five matrix-matrix multiplications with complexity $O(N^3)$, a Randomized-SVD operation with complexity $O(N^2 r)$ to enforce rank constraint (using constant oversampling and power iterations as in [Halko et al., 2011]), and a projection onto $\mathcal{S}$ requiring at most $O(N^2 \log(N))$ for sorting and thresholding operations—across the entire matrix for unstructured sparsity or within blocks for semi-structured sparsity. The overall time complexity is $O(N^3)$.

---

[2]The closed-form solution in Equation (4) can be used to improve other $(\mathbf{S}+\mathbf{L}\mathbf{R})$ methods like HASSLE-free (which employs gradient-descent on a reparameterized $\mathbf{L} = \mathbf{U}\mathbf{V}^\top$)

## 2.3 Convergence of ADMM

Despite its appeal and usage, the convergence properties of 3-Block ADMM remain theoretically challenging. Chen et al. [2016] demonstrated that without additional conditions, the algorithm may fail to converge, while later works [Lin et al., 2015, Wang et al., 2018] established various sufficient conditions for convergence.

We observe that our proposed 3-block ADMM approach can be reformulated as a standard 2-block ADMM by treating $(\mathbf{L}, \mathbf{D})$ as a single variable block. This reformulation is valid because the Lagrangian is separable with respect to $\mathbf{L}$ and $\mathbf{D}$, meaning their joint minimization yields equivalent updates to sequential optimization (although 3-blocks remain the "natural" way to conceptualize the updates). While Meng et al. [2024a] established convergence guarantees for ADMM applied to layerwise pruning, their analysis addresses a different problem formulation than ours. Specifically, they apply ADMM solely to unstructured pruning, whereas our approach extends to $(\mathbf{S} + \mathbf{LR})$ decomposition. Our framework includes a low-rank component with relatively complex updates in each iteration, which introduces additional mathematical challenges in convergence analysis that prevent direct application of the results in Meng et al. [2024a].

To address this gap, we establish the following novel convergence guarantee that ensures the decomposition converges as long as we choose penalty parameter $\rho_t$ that increases sufficiently rapidly (refer to Appendix A for a complete proof).

**Theorem 1.** *Let* $\left\{\mathbf{S}^{(t)}\right\}_{t=0}^{\infty}$ *and* $\left\{\mathbf{L}^{(t)}\right\}_{t=0}^{\infty}$ *be the sequence generated according to update rule* (5). *Suppose the penalty parameter* $\rho_t$ *chosen at iteration* $t$ *is non-decreasing and satisfies* $\sum_{t=0}^{\infty} 1/\rho_t < \infty$. *Then for any* $t \geq 1$:

$$\max\{\|\mathbf{S}^{(t+1)} - \mathbf{S}^{(t)}\|_F, \|\mathbf{L}^{(t+1)} - \mathbf{L}^{(t)}\|_F\} \leq C/\rho_{t-1}, \tag{6}$$

*where* $C$ *is a constant depending on* $\mathbf{X}$, $\widehat{\mathbf{W}}$, $\lambda$, $\rho_0$, *and* $\sum_{t=0}^{\infty} 1/\rho_t$. *In particular, there exists a matrix* $\bar{\mathbf{W}}$ *such that* $\mathbf{S}^{(t)} + \mathbf{L}^{(t)} \to \bar{\mathbf{W}}$ *as* $t \to \infty$.

# 3 Transformer-level matching

After layer-wise pruning, LoRA can directly refine the low-rank components in the $(\mathbf{S} + \mathbf{LR})$ decomposition for task adaptation. However, the sparse components are not well-optimized by this process, as they are determined solely via layer-wise objectives. These layer-wise objectives are imperfect proxies for the true end-to-end loss function. On the other hand, fully optimizing the sparse components using the true end-to-end loss is computationally expensive and requires a full backpropagation on the entire network. To address this limitation, we introduce an efficient *transformer-matching* refinement step that leverages transformer-level information to enhance the sparse components. This procedure is efficient because it requires comparable CUDA memory and runtime to the compression algorithms themselves.

Our transformer-matching procedure jointly optimizes all sparse and low-rank components across layers within a transformer block to better match the original transformer's output. It acts as an intermediate loss function between layer-wise proxies and the true end-to-end loss. This approach can enhance any $(\mathbf{S} + \mathbf{LR})$ decomposition, including pruning (where $\mathbf{LR} = \mathbf{0}$). Figure 2 illustrates the performance gains obtained after applying `TM` to state-of-the-art one-shot $(\mathbf{S} + \mathbf{LR})$ decomposition algorithms. In Table 3, we show results of applying `transformer-matching` to pruning algorithms with pure sparsity constraints like WandA [Sun et al., 2024], SparseGPT [Frantar and Alistarh, 2023], and ALPS [Meng et al., 2024a] highlighted in dark red.

Formally, for each transformer block $T_i$ with $L$ layers, after obtaining sparse and low-rank components $\{\mathbf{S}^{(i,\ell)}, \mathbf{L}^{(i,\ell)}\}_{\ell=1}^{L}$ through layer-wise pruning, we denote the support of sparse components as $\mathcal{S}^{(i,\ell)} = \text{Supp}(\mathbf{S}^{(i,\ell)})$. Let $\mathbf{X}_i$ represent the outputs from the previously compressed transformer block $T_{i-1}$. We then refine these components using a transformer-level reconstruction loss:

$$\min_{\{\mathbf{S}^{(i,\ell)}, \mathbf{L}^{(i,\ell)}\}_{\ell=1}^{L}} \left\| T_i\left(\mathbf{X}_i; \{\mathbf{W}^{(i,\ell)}\}_{\ell=1}^{L}\right) - T_i\left(\mathbf{X}_i; \{\mathbf{S}^{(i,\ell)} + \mathbf{L}^{(i,\ell)}\}_{\ell=1}^{L}\right) \right\|_F^2,$$

$$\text{s.t.} \quad \text{Supp}(\mathbf{S}^{(i,\ell)}) \subset \mathcal{S}^{(i,\ell)}, \quad \text{rank}(\mathbf{L}^{(i,\ell)}) \leq r^{(i,\ell)} \tag{7}$$

where this constraint optimizes the weights of the decomposed components. Due to the non-linear activations between layers, we use gradient-based optimization methods such as Adam. Nonetheless, this optimization remains computationally efficient as it is performed using iteratively chunks

of the small calibration dataset used for compression. Additionally, the forward/backward passes are limited to only one transformer block. The transformer-matching approach offers two key advantages. First, it creates a more accurate proxy of the original loss function by directly minimizing the discrepancy between the original and compressed transformer outputs, resulting in higher-performance pruned models. Second, it reduces accumulated errors—introduced in layer-wise pruning where input activations $\mathbf{X}$ are computed from outputs of previously *pruned* layers—by ensuring that activations fed into subsequent layers more faithfully match those of the dense model:

$$T_i \left( \mathbf{X}_i; \{\mathbf{S}^{(i,\ell)} + \mathbf{L}^{(i,\ell)}\}_{\ell=1}^L \right) = \mathbf{X}_{i+1} \approx \mathbf{X}_{i+1}^{(\text{oracle})} = T_i \left( \mathbf{X}_i; \{\mathbf{W}^{(i,\ell)}\}_{\ell=1}^L \right), \qquad (8)$$

therefore providing better activation statistics for compression on subsequent transformers compared to layer-wise reconstruction which only matches weight matrices layer by layer.

After transformer-matching, the refined sparse components $\mathbf{S}^{(i,\ell)}$ remain fixed during downstream fine-tuning, while the low-rank components $\mathbf{L}^{(i,\ell)}$ serve as smart initializations for efficient LoRA adaptation to specific tasks.

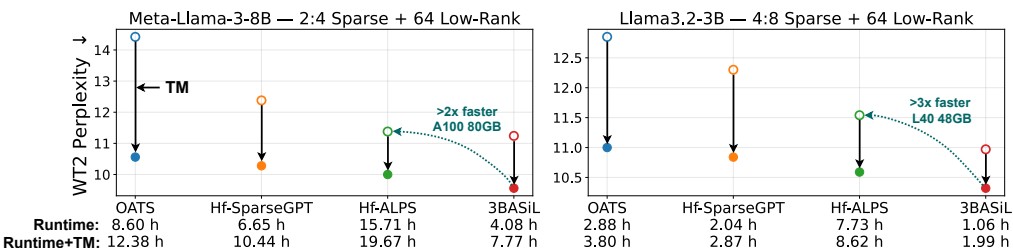

Figure 2: Our transformer matching (TM) procedure improves any one-shot ($\mathbf{S} + \mathbf{LR}$) decomposition method (see baselines in Section 4) with a small computational overhead. Circled markers represent standard ($\mathbf{S}+\mathbf{LR}$) methods, while filled markers indicate their TM-enhanced versions. Black arrows illustrate performance gains due to TM. The compression runtimes are reported in hours. Llama3-8B models were run on a A100 GPU, while Llama3.2-3B were run on a L40 GPU. Our proposal 3BASiL-TM, remains significantly faster: **(left)** over $2\times$ speedup on an A100 80GB for the Llama3-8B model decomposed to (2:4+64LR) configuration, and **(right)** over $3\times$ speedup on an L40 48GB for the Llama3.2-3B model decomposed to (4:8+64LR) configuration (both compared to Hf-ALPS).

## 4 Experimental results

### 4.1 Experimental setup

**Models and LLM Evaluation Protocol**  To rigorously assess the effectiveness of our proposed approach 3BASiL and *transformer-matching* (TM) procedure, we conducted extensive experiments on the Llama-3 and Llama-3.2 model families [Dubey et al., 2024] and scaled results in one experiment to a OPT-30B [Zhang et al., 2022] model, hence covering architectures with number of parameters ranging from 1B to 30B. Following the widely adopted setup introduced by Frantar and Alistarh [2023] for one-shot compression, we select the calibration set consisting of 128 randomly sampled text segments (2048 tokens each) from the C4 [Raffel et al., 2020] train dataset's first shard. This calibration set is shared across all evaluated compression methods to ensure consistency.

We adopt two evaluation criteria: (1) *perplexity* as a foundational measure of language modeling quality, and (2) *zero-shot task performance* to assess practical downstream capabilities post-compression. Perplexity is measured using three standard benchmarks: WikiText2 [Merity et al., 2017], Penn Treebank [Marcus et al., 1994], and C4 validation samples, computed using Hugging-Face's full-stride perplexity protocol [Per, 2022]. For zero-shot evaluation, we utilize the LM Harness framework [Gao et al.] on a diverse suite of eight zero-shot tasks: PIQA [Bisk et al., 2020], ARC-Easy/Challenge [Clark et al., 2018], HellaSwag [Zellers et al., 2019], Winogrande [Sakaguchi et al., 2021], RTE [Poliak, 2020], OpenbookQA [Banerjee et al., 2019], and BoolQ [Clark et al., 2019]. We report individual scores for each benchmark and the average across all tasks.

For perplexity, ($\downarrow$) lower values are preferred. For zero-shot tasks, ($\uparrow$) higher values are preferred.

**Baselines**  Our main baselines are OATS [Zhang and Papyan, 2025], HASSLE-free-SparseGPT (Hf-SparseGPT) and HASSLE-free-ALPS (Hf-ALPS)—the latter two use pruning approaches

SparseGPT [Frantar and Alistarh, 2023] and ALPS [Meng et al., 2024a], respectively, in the sparsification step of the alternating minimization algorithm proposed by Makni et al. [2025].

For all these baselines, we follow the original configuration and perform 80 steps of alternating minimization. For HASSLE-free methods, we propose an improved implementation that replaces their original parameterization of $\mathbf{L} = \mathbf{U}\mathbf{V}^\top$ and gradient-based optimization with the closed-form solution provided in Equation (4). This modification leads to improved compression runtime and better downstream LLM evaluation metrics–see Table 6. Under this improved implementation, the method EoRA [Liu et al., 2024], which applies the update in Equation (4) once after one round of compression, reduces to HASSLE-free (alternating minimization approach) with a number of iterations equal to one. EoRA is the fastest $(\mathbf{S} + \mathbf{LR})$ method but underperforms HASSLE-free which uses more alternating minimization steps (default=80), and hence there is a large gap compared to our approach on most model/configuration settings. We show some results of EoRA in Table 6.

More details on the implementation of 3BASiL, TM and the baselines (with improved implementation) are provided in Appendix B.

## 4.2 Numerical results

Our evaluation focuses primarily on $(\text{N:M} + \mathbf{LR})$ decompositions, which enable efficient GPU acceleration via specialized CUDA kernels [Mozaffari et al., 2024, Makni et al., 2025]. We evaluate both one-shot compression performance and downstream LoRA fine-tuning capabilities. Additionally, we demonstrate the generality of our approach through experiments with unstructured sparsity and integration with sparsity allocation methods. The downstream LoRA experiments have been motivated by recent studies [Li et al., 2024, Guo et al., 2024, Saha et al., 2024] suggesting that decompositions of the form $\mathcal{C}(\mathbf{W}) + \mathbf{LR}$ are LoRA-aware: i.e. low-rank components obtained from compression can act as *smart initialization* to improve downstream LoRA fine-tuning. Further numerical experiments where we ablate on TM and LoRA fine-tuning for $(\mathbf{S} + \mathbf{LR})$ methods can be found in Appendix C.

**One-shot (Sparse + LR) results** We compare 3BASiL to prior $(\mathbf{S} + \mathbf{LR})$ decomposition methods in the one-shot compression setting—i.e., without fine-tuning. Table 1 reports results for LLaMA3.2 family under various (N:M + 64LR) configurations. Table 2 and Figure 3 show results for similar configurations for the LLaMA3-8B model. 3BASiL reduces perplexity by up to 8% compared to previous SOTA (due to better layer-wise reconstruction—see Figure 5a and Figure 5b), with the TM step yielding further dramatic improvements of up to 40% perplexity reduction.

We also compare $(\mathbf{S} + \mathbf{LR})$ decompositions with semi-structured pure pruning methods under a fixed compression ratio $\rho = 50\%$. Results in Table 3 show that 3BASiL-TM achieves the best compression-performance trade-off under $(3:8 + \text{LR})$ configurations among different $(\mathbf{S} + \mathbf{LR})$ methods. Additionally, we expand our $(\mathbf{S} + \mathbf{LR})$ experiments to include a $(2:4 + 112)$ configuration for OPT-30B model [Zhang et al., 2022]. This configuration uses a 1.56% Low-Rank Adapter (hidden size 7168). Under this configuration, Mozaffari et al. [2024] report a 1.53x speedup as well as a 0.63x memory reduction compared to dense model. Results are reported in Table 4.

For unstructured sparsity configurations, we benchmark 3BASiL against prior $(\mathbf{S} + \mathbf{LR})$ methods on a "less aggressive" (50% + 128) compression for both Llama3.2-1B and Llama3-8B models. Table 5 shows that our proposed method maintains its advantage even in this near-lossless configuration regime. We further evaluate 3BASiL under high sparsity ratios with (Unstructured + 64) configurations and demonstrate how our method integrates with the sparsity allocation method OWL [Yin et al., 2024] for the Llama3-8B model—see Table 13 in Appendix B.

These results highlight the effectiveness and flexibility of our method 3BASiL.

**LoRA fine-tuning after one-shot compression** After applying $(\mathbf{S} + \mathbf{LR})$ decomposition, the resulting low-rank components can serve as initialization for LoRA fine-tuning on downstream tasks to recover lost performance. We conducted limited LoRA fine-tuning on 10% of the first C4 training dataset shard (approximately 15 million tokens), with detailed hyperparameters in Appendix B. Figure 4a demonstrates that LFT-3BASiL-TM significantly reduces the C4 perplexity of $(\mathbf{S}+\mathbf{LR})$ decompositions, particularly under aggressive compression regimes like 2:8+64LR. Moreover, while LoRA fine-tuning can recover a large portion of the performance lost due to compression, an advanced one-shot decomposition approach retains its advantage post fine-tuning. For instance, LFT-3BASiL-TM still outperforms competing decomposition methods after LoRA fine-tuning of 2:8+64LR configurations, achieving approximately 8% lower perplexity.

Figure 3: One-shot C4 perplexity analysis of Llama3-8B under different (N:M + 64LR) configurations.

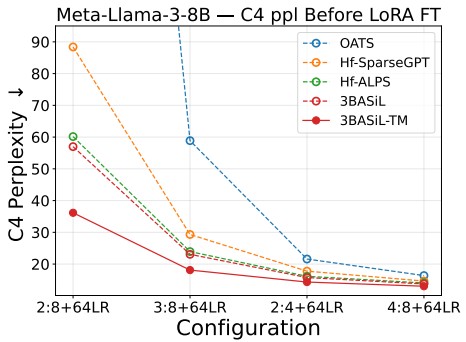

Table 1: Perplexity of Llama-3.2 family

| Method | Config | Llama-3.2-1B | | | Llama-3.2-3B | | |
|---|---|---|---|---|---|---|---|
| | | C4 ↓ | WT2 ↓ | PTB ↓ | C4 ↓ | WT2 ↓ | PTB ↓ |
| OATS | 2:8+64LR | 640.86 | 605.20 | 779.86 | 531.47 | 494.31 | 674.71 |
| Hf-SparseGPT | | 162.45 | 134.21 | 170.12 | 106.07 | 106.17 | 151.92 |
| Hf-ALPS | | 107.14 | 94.71 | 124.17 | 69.96 | 65.34 | 108.68 |
| 3BASiL | | 97.50 | 86.59 | 100.35 | 73.00 | 72.26 | 110.10 |
| 3BASiL-TM | | **55.24** | **49.74** | **69.49** | **45.35** | **42.38** | **68.29** |
| OATS | 3:8+64LR | 125.91 | 92.13 | 115.80 | 65.08 | 47.27 | 81.29 |
| Hf-SparseGPT | | 43.50 | 34.18 | 51.16 | 34.66 | 26.60 | 39.76 |
| Hf-ALPS | | 37.80 | 29.00 | 43.60 | 27.94 | 22.77 | 34.59 |
| 3BASiL | | 34.81 | 26.96 | 41.55 | 26.35 | 20.66 | 31.77 |
| 3BASiL-TM | | **26.26** | **20.75** | **32.09** | **20.89** | **17.18** | **25.31** |
| OATS | 4:8+64LR | 28.06 | 19.69 | 32.90 | 19.25 | 13.40 | 21.67 |
| Hf-SparseGPT | | 22.24 | 15.90 | 27.35 | 17.09 | 12.30 | 19.19 |
| Hf-ALPS | | 20.71 | 14.90 | 24.75 | 16.04 | 11.51 | 18.17 |
| 3BASiL | | 20.04 | 14.26 | 24.27 | 15.65 | 10.97 | 17.39 |
| 3BASiL-TM | | **18.66** | **13.19** | **22.46** | **14.89** | **10.29** | **16.52** |
| OATS | 2:4+64LR | 41.80 | 28.45 | 45.36 | 25.18 | 17.41 | 28.60 |
| Hf-SparseGPT | | 27.25 | 19.45 | 32.63 | 20.38 | 15.03 | 23.23 |
| Hf-ALPS | | 23.90 | 17.66 | 28.96 | 18.45 | 13.79 | 20.50 |
| 3BASiL | | 23.16 | 17.27 | 27.77 | 17.89 | 13.12 | 20.10 |
| 3BASiL-TM | | **20.46** | **15.23** | **24.60** | **16.37** | **11.79** | **18.34** |
| **Dense** | – | 14.01 | 9.75 | 17.59 | 11.33 | 7.81 | 13.53 |

Table 2: One-shot (N:M Sparse + LR) decomposition performance for Meta-Llama-3-8B.

| Method | Config | C4 ↓ | WT2 ↓ | PTB ↓ | PIQA ↑ | HS ↑ | ARC-E ↑ | ARC-C ↑ | WG ↑ | RTE ↑ | OQA ↑ | BoolQ ↑ | Avg ↑ |
|---|---|---|---|---|---|---|---|---|---|---|---|---|---|
| OATS | 3:8+64LR | 58.88 | 40.76 | 67.35 | 63.71 | 39.48 | 42.68 | 24.32 | 53.91 | 52.71 | 28.40 | 63.98 | 46.15 |
| Hassle-free-SparseGPT | | 29.32 | 21.46 | 32.06 | 68.66 | 51.99 | 50.97 | 30.38 | 63.85 | 53.07 | 32.00 | 71.31 | 52.78 |
| Hassle-free-ALPS | | 23.93 | 18.20 | 26.31 | 70.62 | 56.54 | 54.42 | 30.12 | 64.72 | 55.23 | 32.80 | 71.96 | 54.55 |
| 3BASiL | | 23.07 | 18.03 | 24.84 | 71.06 | 56.96 | 57.70 | 32.59 | 66.69 | 54.51 | 33.00 | 66.70 | 54.90 |
| 3BASiL-TM | | **18.11** | **14.26** | **20.47** | **74.05** | **61.85** | **60.73** | **34.73** | 65.98 | 54.51 | **34.80** | **76.91** | **57.94** |
| OATS | 4:8+64LR | 16.38 | 10.88 | 17.23 | 75.84 | 67.60 | 67.09 | 41.21 | 70.88 | 60.29 | 38.20 | 73.61 | 61.84 |
| Hassle-free-SparseGPT | | 14.65 | 9.88 | 15.21 | 77.09 | 69.95 | 69.32 | 41.81 | 71.27 | 56.32 | 40.60 | 79.39 | 63.22 |
| Hassle-free-ALPS | | 14.04 | 9.44 | 14.45 | 76.82 | 71.19 | 71.04 | 44.45 | **72.77** | 56.68 | 40.20 | 78.13 | 63.91 |
| 3BASiL | | 13.74 | 9.21 | 14.24 | 76.88 | 72.05 | 70.16 | 44.80 | 72.14 | 61.01 | 41.40 | 80.89 | 64.92 |
| 3BASiL-TM | | **13.02** | **8.64** | **13.70** | **78.24** | **72.59** | **73.11** | **47.35** | 71.98 | **63.18** | **42.40** | 80.49 | **66.17** |
| OATS | 2:4+64LR | 21.59 | 14.76 | 23.41 | 72.74 | 60.70 | 60.86 | 34.81 | 65.51 | 57.76 | 35.20 | 68.32 | 56.99 |
| Hassle-free-SparseGPT | | 17.77 | 12.38 | 18.71 | 74.81 | 65.04 | 66.16 | 38.57 | 70.09 | 54.87 | 38.40 | 77.71 | 60.71 |
| Hassle-free-ALPS | | 16.15 | 11.38 | 16.71 | 75.19 | 67.10 | 64.44 | 38.91 | 69.53 | 59.93 | 39.40 | 78.38 | 61.61 |
| 3BASiL | | 15.76 | 11.23 | 16.25 | 76.50 | 67.61 | 67.21 | 40.10 | 70.24 | 64.26 | 38.20 | 78.29 | 62.80 |
| 3BASiL-TM | | **14.34** | **9.78** | **14.88** | **77.48** | **69.58** | 67.21 | **40.53** | **71.27** | 61.37 | **39.80** | **79.51** | **63.34** |
| **Meta-Llama-3-8B Dense** | – | 9.44 | 6.14 | 11.18 | 80.79 | 79.17 | 77.69 | 53.33 | 72.85 | 69.68 | 45.00 | 81.44 | 69.99 |

| Method | Config | C4 ↓ | WT2 ↓ | PTB ↓ | PIQA ↑ | ARC-E ↑ | ARC-C ↑ |
|---|---|---|---|---|---|---|---|
| Wanda | 2:4 | 38.21 | 26.89 | 47.13 | 67.63 | 49.37 | 29.01 |
| Wanda-TM | | 15.91 | 11.03 | 17.60 | 75.14 | 63.97 | 40.19 |
| SparseGPT | | 22.65 | 16.22 | 25.15 | 71.16 | 56.48 | 32.59 |
| SparseGPT-TM | | 15.30 | 10.83 | 16.77 | 76.28 | 65.28 | 40.36 |
| ALPS | | 19.62 | 14.50 | 21.73 | 73.78 | 60.06 | 35.84 |
| ALPS-TM | | 14.96 | 10.65 | 16.35 | 76.88 | 65.03 | 39.85 |
| Wanda | 4:8 | 22.70 | 15.58 | 26.62 | 72.03 | 58.63 | 36.09 |
| Wanda-TM | | 13.99 | 9.28 | 15.09 | 77.48 | 68.22 | 42.75 |
| SparseGPT | | 17.59 | 12.29 | 18.48 | 75.68 | 63.38 | 39.59 |
| SparseGPT-TM | | 13.68 | 9.28 | 14.51 | 78.07 | 70.29 | 43.94 |
| ALPS | | 16.06 | 11.17 | 16.60 | 76.12 | 66.25 | 40.87 |
| ALPS-TM | | 13.59 | 9.15 | 14.18 | 77.58 | 69.57 | 43.94 |
| OATS | 2:8+LR | 21.03 | 14.54 | 24.15 | 73.67 | 59.68 | 37.12 |
| Hf-SparseGPT | | 20.05 | 15.03 | 22.01 | 74.05 | 60.52 | 36.18 |
| Hf-ALPS | | 17.89 | 13.07 | 19.11 | 74.54 | 65.53 | 39.08 |
| 3BASiL | | 15.20 | 10.64 | 15.80 | 76.71 | 70.08 | 43.52 |
| 3BASiL-TM | | 13.81 | 9.50 | 14.74 | 77.15 | 73.36 | 44.54 |
| OATS | 3:8+LR | 16.87 | 11.43 | 18.53 | 75.24 | 65.91 | 39.85 |
| Hf-SparseGPT | | 16.16 | 11.36 | 16.71 | 75.79 | 67.55 | 41.04 |
| Hf-ALPS | | 14.85 | 10.20 | 15.42 | 77.15 | 69.40 | 43.64 |
| 3BASiL | | 13.73 | 9.29 | 14.62 | **78.45** | 71.42 | 43.43 |
| 3BASiL-TM | | 13.01 | 8.69 | 13.74 | 77.80 | **75.00** | **47.44** |
| **Llama3-8B Dense** | – | 9.44 | 6.14 | 11.18 | 80.79 | 77.69 | 53.33 |

Table 3: One-shot (N:M Sparse + LR) decomposition performance of Llama3-8B model. The compression ratio (percentage of non-zero parameters retained) is fixed to be $\rho = 0.5$. For Perplexity, (↓) lower values are preferred. For zero-shot tasks, (↑) higher values are preferred. Bolded values correspond to the overall best compression scheme that satisfies $\rho = 0.5$. Underlined values correspond to the best pure pruning algorithm for the same compression. This shows the universality of `transformer-matching` to pure sparsity constraints.

| Method | C4 ↓ | WT2 ↓ | PTB ↓ | Time (hrs) ↓ |
|---|---|---|---|---|
| OATS-10 | 11.75 | 10.48 | 14.65 | 5.81 |
| Hf-SparseGPT | 11.58 | 10.17 | 14.39 | 5.97 |
| Hf-ALPS-10 | 11.56 | 10.05 | 14.33 | 4.33 |
| 3BASiL | **11.53** | **10.04** | **14.26** | **4.20** |
| **Dense** | 11.44 | 9.56 | 14.04 | – |

Table 4: One-shot (2:4 + 112) decomposition of OPT-30B model. This configuration results in efficient inference. We limit the compression runtime to 6 A100 GPU hours. `3BASiL-TM` largely exceeds this period. We limit the alternating minimization steps of `Hf-ALPS` and `OATS` to 10 to fit within the time constraint.

Table 5: One-shot $(50\% + 128)$ decomposition for Llama3.2-1B and Meta-Llama-3-8B models.

| Method | Config | C4↓ | WT2↓ | PTB↓ | PIQA↑ | HS↑ | ARC-E↑ | ARC-C↑ | WG↑ | RTE↑ | OQA↑ | BoolQ↑ | Avg↑ |
|---|---|---|---|---|---|---|---|---|---|---|---|---|---|
| OATS | | 17.99 | 12.16 | 21.40 | 71.71 | 57.96 | 57.28 | 33.79 | 59.98 | 52.71 | 33.00 | 63.94 | 53.80 |
| Hf-SparseGPT | | 17.25 | 11.99 | 20.87 | 72.91 | 59.04 | 56.82 | 33.11 | 58.88 | 57.76 | 35.00 | 57.03 | 53.82 |
| Hf-ALPS | 50%+128 | 16.81 | 11.66 | 20.12 | 72.80 | 59.92 | 57.62 | 33.11 | 58.64 | 55.96 | 35.20 | 59.66 | 54.11 |
| 3BASiL | | 16.17 | 11.16 | 20.00 | **73.83** | 60.42 | 58.04 | 34.47 | 60.38 | 53.79 | **36.80** | 58.20 | 54.49 |
| 3BASiL-TM | | **15.78** | **10.87** | **19.33** | 73.23 | **60.66** | **59.26** | **34.56** | **61.01** | **59.21** | 36.60 | **64.13** | **56.08** |
| **Llama-3.2-1B Dense** | – | 14.01 | 9.75 | 17.59 | 74.59 | 63.66 | 60.48 | 36.26 | 60.69 | 56.68 | 37.20 | 63.98 | 56.69 |
| OATS | | 12.25 | 7.78 | 12.92 | 78.40 | 75.32 | 73.99 | 49.15 | **73.80** | 58.84 | 41.80 | 79.42 | 66.34 |
| Hf-SparseGPT | | 11.98 | 7.77 | 12.85 | 79.11 | 75.88 | 75.00 | 49.40 | 73.32 | 63.18 | 43.80 | 78.32 | 67.25 |
| Hf-ALPS | 50%+128 | 12.09 | 7.99 | 12.86 | 78.78 | 76.29 | **76.52** | **51.19** | 73.09 | 60.65 | 40.20 | **81.62** | 67.29 |
| 3BASiL | | 11.51 | 7.47 | 12.36 | 79.54 | **76.69** | 74.75 | 48.72 | 72.69 | 67.87 | 43.00 | 80.24 | 67.94 |
| 3BASiL-TM | | **11.27** | **7.30** | **12.26** | **79.65** | 76.07 | 75.84 | 47.78 | 71.98 | **70.40** | **44.20** | 80.70 | **68.33** |
| **Meta-Llama-3-8B Dense** | – | 9.44 | 6.14 | 11.18 | 80.79 | 79.17 | 77.69 | 53.33 | 72.85 | 69.68 | 45.00 | 81.44 | 69.99 |

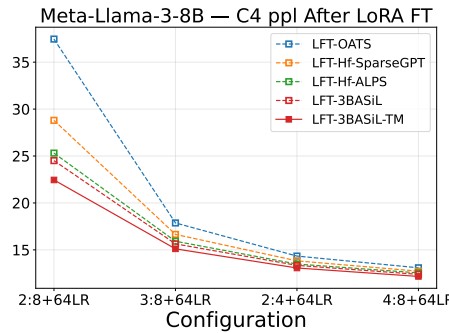

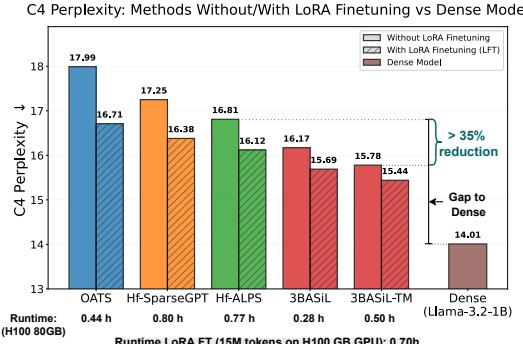

(a) C4 ppl of Llama3-8B model under different $(\mathbf{S} + \mathbf{LR})$ configurations after LoRA.

(b) C4 perplexity gap to dense model (Llama3.2-1B) under (50%+128LR) configuration.

Figure 4: C4 perplexity performance of Llama3-8B & Llama3.2-1B before/after LoRA fine-tuning.

# 5   Related Work

**One-shot Sparse/Quantized plus Low-Rank compression**   The seminal works of Yu et al. [2017] proposed a compression technique for a neural network using sparsity plus low-rank constraints. However, the authors study small-scale vision models. In addition, they consider a compression that needs to be repeated over multiple rounds (decomposing selected layers and followed by a re-training process). Our focus is different; we are interested in compressing at LLM-scale in one-shot (no expensive retraining). Recent methods in LLM compression have focused on effectively combining low-rank decomposition with quantization or sparsity. EoRA [Liu et al., 2024] has been proposed as a method to compensate for the loss produced by a general-purpose compressed weight $\mathcal{C}(\mathbf{W})$ using a low-rank component, it does the low-rank fitting step once post the initial weight compression, which could include combinations of sparsity and quantization. LoftQ [Li et al., 2024] jointly optimizes quantization and LoRA initialization by solving $\min_{\mathbf{Q},\mathbf{L}} \|\mathbf{W} - (\mathbf{Q}+\mathbf{L})\|_F$, where $\mathbf{W}$ represents the original weights, $\mathbf{Q}$ the quantized component, and $\mathbf{L}$ the low-rank component. LQ-LoRA [Guo et al., 2024] extends this by incorporating Fisher information weighting, approximately solving $\min_{\mathbf{Q},\mathbf{L}} \|\mathbf{F} \odot (\mathbf{W} - (\mathbf{Q}+\mathbf{L}))\|_F$. CALDERA [Saha et al., 2024] further considers the layer-wise reconstruction error, optimizing $\min_{\mathbf{Q},\mathbf{L}} \|\mathbf{XW} - \mathbf{X}(\mathbf{Q}+\mathbf{L})\|_F$ to maintain the outputs of individual layers rather than mere weight approximation. From the $(\mathbf{S} + \mathbf{LR})$ perspective, OATS [Zhang and Papyan, 2025] proposes an outlier-aware alternating minimization, effectively reducing to solving $\min_{\mathbf{S},\mathbf{L}} \|\mathbf{DW} - \mathbf{D}(\mathbf{S}+\mathbf{L})\|_F$ with $\mathbf{D} = \mathrm{diag}(\mathbf{X}^T\mathbf{X})$, as noted by Makni et al. [2025]. HASSLE-free [Makni et al., 2025] directly tackles layer-wise reconstruction error $\min_{\mathbf{S},\mathbf{L}} \|\mathbf{XW} - \mathbf{X}(\mathbf{S}+\mathbf{L})\|_F$ using alternating minimization. While methods such as OATS and HASSLE-free separately optimize sparse and low-rank components, our proposed approach, 3BASiL, distinctly utilizes a unified optimization framework via a 3-block ADMM formulation, jointly optimizing sparse and low-rank components simultaneously.

**Sparse plus Low-Rank structures in transformers**   Beyond *model compression*, sparse plus low-rank structures have a strong presence in the context of LLMs. LoRAPrune [Zhang et al., 2024] is a purse sparsification method, which prunes a model (iteratively) by designing a memory-efficient LoRA-guided (low-rank structure) pruning criterion. In contrast, LoSA (low-rank Sparse Adapta-

tion) [Huang et al., 2025] jointly applies LoRA fine-tuning and pruning in a unified framework to obtain a fine-tuned sparse-only (as opposed to $(\mathbf{S} + \mathbf{LR})$)) model, by dynamically sparsifying the LoRA weights and adjusting their rank. SLTrain [Han et al., 2024] addresses $(\mathbf{S} + \mathbf{LR})$ from a training perspective. It pre-trains an LLM using a fixed random sparse mask plus trainable low-rank factors (similar to LoRA), achieving comparable accuracy to dense training with far fewer parameters. SLTrain demonstrates the benefits of $(\mathbf{S} + \mathbf{LR})$ structure for pre-training but it doesn't solve the post-hoc decomposition problem of a dense model. There are connections between our *transformer-matching* step and SLTrain as they both train sparse (fixed support) and low-rank components, but they minimize different loss functions and serve different purposes.

**ADMM approaches to compress networks**    The Alternating Direction Method of Multipliers (ADMM) [Boyd et al., 2011, Davis and Yin, 2016] is an effective optimization technique for problems with coupled variables that has been successfully applied to neural network compression. Ye et al. [2018] introduced ADMM-based progressive weight pruning that optimizes the original loss function under sparsity constraints, which Ye et al. [2019] extended to preserve adversarial robustness during compression. In contrast, recent methods have scaled ADMM to LLMs through layer-wise reconstruction: Boža [2024] employed ADMM to solve a convex problem recovering optimal weights on a fixed support of the weight matrix, while Meng et al. [2024a] utilized ADMM for a non-convex problem that jointly optimizes both support and weights. Our proposed method differs from these prior works as we explore a 3-block ADMM in model compression that simultaneously optimizes $(\mathbf{S} + \mathbf{LR})$ components with theoretical convergence guarantees.

**Exact Low-Rank updates for layer-wise compression**    The problem of exact low-rank updates found in Equation (4) has original roots from classical reduced-rank regression methods [Izenman, 1975, Reinsel and Velu, 1998], which provide closed-form solutions for optimally approximating linear regression models under rank constraints. Recent work, including CALDERA [Saha et al., 2024] and the low-rank correction method by Scetbon and Hensman [2024], applies these closed-form updates to compress large language models into $\mathbf{W} \approx \mathbf{Q} + \mathbf{LR}$. We also use these exact low-rank updates by integrating them directly in Equation (4) within our ADMM framework for $(\mathbf{S} + \mathbf{LR})$ decomposition.

# 6    Conclusion and limitations

We present `3BASiL` as a highly-efficient $(\mathbf{S} + \mathbf{LR})$ decomposition algorithm with theoretical convergence guarantees. It provides high-quality solutions to the layer-wise decomposition problem presented in Equation (1) in terms of objective minimization (Figure 5a and Figure 5b) compared to competing $(\mathbf{S} + \mathbf{LR})$ decomposition methods. We further refine these decomposed weights with our novel (memory-efficient) transformer matching step `TM` that can enhance any $(\mathbf{S} + \mathbf{LR})$ decomposition. This shows that one route for optimal compression results (in the context of $\mathcal{C}(\mathbf{W}) + \mathbf{LR}$) is to unfold the LLM compression into 3 minimization steps: (i) [layer-wise reconstruction] this is the loss that has been considered in many SOTA pruning/quantization algorithms [Frantar and Alistarh, 2023, Meng et al., 2024a, Saha et al., 2024, Frantar et al., 2022, Meng et al., 2024b], (ii) [transformer-matching] this is an intermediate loss function (to be optimized in a memory-efficient manner) which is a more reliable approximation to the true loss function than simple layer-wise reconstruction, and (iii) [LoRA fine-tuning] plugs the obtained low-rank components as *smart initialization* for LoRA to minimize the true LLM loss function. We believe that our 3-block ADMM approach and `TM` can generalize to quantization or quantized-sparse constraints. We leave these explorations for future works.While we have shown how to integrate sparsity allocation mechanisms like OWL to our framework, it remains to explore dedicated methods that can algorithmically allocate different sparsity/rank configurations to different layers to further improve efficiency-utility-computations tradeoffs.

**Acknowledgements**

This research is supported in part by grants from the Office of Naval Research (N000142512504, N000142212665). We acknowledge the MIT Engaging cluster for providing HPC resources that have contributed to the research results reported within this paper. Additionally, we thank Google for providing us with Google Cloud Credits. We thank Shibal Ibrahim, Ryan Lucas, and Gabriel Afriat for their helpful discussions.

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

# Appendix

## A Proofs of Theorem 1

*Proof.* For conciseness, throughout the proof, we denote $\mathbf{H} = \mathbf{X}^{\top}\mathbf{X} + \lambda\mathbf{I}$ and $\mathbf{G} = \left(\mathbf{X}^{\top}\mathbf{X} + \lambda\mathbf{I}\right)\widehat{\mathbf{W}}$. We denote $C_F$ as a large constant such that

$$\max\{1, \|\mathbf{H}^{-1/2}\|_2, \|\mathbf{H}\|_2, \|\mathbf{G}\|_F\} \leq C_F. \tag{9}$$

To establish the theorem, we first present the following three lemmas.

**Lemma A.1.** *Let $\left\{\mathbf{D}^{(t)}\right\}_{t=0}^{\infty}$, $\left\{\mathbf{L}^{(t)}\right\}_{t=0}^{\infty}$ and $\left\{\mathbf{V}^{(t)}\right\}_{t=0}^{\infty}$ be the sequence generated according to update rule (5). Then for any $t \geq 1$, it holds*

$$\|\mathbf{L}^{(t)}\|_F \leq C_F^3 \left(1 + \|\mathbf{D}^{(t)}\|_F + \frac{\|\mathbf{V}^{(t)}\|_F}{\rho_{t-1}} + \frac{\|\mathbf{V}^{(t-1)}\|_F}{\rho_{t-1}}\right). \tag{10}$$

**Lemma A.2.** *Let $\left\{\mathbf{D}^{(t)}\right\}_{t=0}^{\infty}$ and $\left\{\mathbf{V}^{(t)}\right\}_{t=0}^{\infty}$ be the sequence generated according to update rule (5). Then for any $t \geq 1$, it holds*

$$\|\mathbf{V}^{(t+1)}\|_F \leq (C_F + C_F^4) \left(1 + \|\mathbf{D}^{(t)}\|_F + \frac{\|\mathbf{V}^{(t)}\|_F}{\rho_{t-1}} + \frac{\|\mathbf{V}^{(t-1)}\|_F}{\rho_{t-1}}\right). \tag{11}$$

*and*

$$\|\mathbf{D}^{(t+1)} - \mathbf{D}^{(t)}\|_F \leq \frac{2C_F + 2C_F^4}{\rho_t} \left(1 + \|\mathbf{D}^{(t)}\|_F + \frac{\|\mathbf{V}^{(t)}\|_F}{\rho_{t-1}} + \frac{\|\mathbf{V}^{(t-1)}\|_F}{\rho_{t-1}}\right). \tag{12}$$

**Lemma A.3.** *Let $\left\{\mathbf{D}^{(t)}\right\}_{t=0}^{\infty}$ and $\left\{\mathbf{V}^{(t)}\right\}_{t=0}^{\infty}$ be the sequence generated according to update rule (5). Then for any $t \geq 1$, it holds*

$$
\begin{aligned}
&\|\mathbf{D}^{(t)}\|_F + \frac{\|\mathbf{V}^{(t)}\|_F}{\rho_{t-1}} + \frac{\|\mathbf{V}^{(t-1)}\|_F}{\rho_{t-1}} \\
&\leq \exp\left(3(C_F + C_F^4)\sum_{s=1}^{t-1}\frac{1}{\rho_{s-1}}\right) \cdot \left(\|\mathbf{D}^{(1)}\|_F + \frac{\|\mathbf{V}^{(1)}\|_F}{\rho_0} + \frac{\|\mathbf{V}^{(0)}\|_F}{\rho_0} + \sum_{s=1}^{t-1}\frac{3(C_F + C_F^4)}{\rho_{s-1}}\right)
\end{aligned}
\tag{13}
$$

Returning to the proof of the main theorem, define

$$
\begin{aligned}
C_A = 2(C_F + C_F^4)\Bigg[&1 + \exp\left(3(C_F + C_F^4)\sum_{s=1}^{\infty}\frac{1}{\rho_{s-1}}\right) \cdot \\
&\left(\|\mathbf{D}^{(1)}\|_F + \frac{\|\mathbf{V}^{(1)}\|_F}{\rho_0} + \frac{\|\mathbf{V}^{(0)}\|_F}{\rho_0} + \sum_{s=1}^{\infty}\frac{3(C_F + C_F^4)}{\rho_{s-1}}\right)\Bigg].
\end{aligned}
\tag{14}
$$

It follows from the update rules (5) that $C_A$ is a constant depending on $\mathbf{X}, \widehat{\mathbf{W}}, \lambda, \rho_0,$ and $\sum_{t=0}^{\infty}1/\rho_t$.

Lemma A.2 together with Lemma A.3 yields

$$\|\mathbf{D}^{(t+1)} - \mathbf{D}^{(t)}\|_F \leq \frac{2C_F + 2C_F^4}{\rho_t}\left(1 + \|\mathbf{D}^{(t)}\|_F + \frac{\|\mathbf{V}^{(t)}\|_F}{\rho_{t-1}} + \frac{\|\mathbf{V}^{(t-1)}\|_F}{\rho_{t-1}}\right) \leq \frac{C_A}{\rho_t}. \tag{15}$$

and

$$\|\mathbf{V}^{(t+1)}\|_F \leq (C_F + 2C_F^4)\left(1 + \|\mathbf{D}^{(t)}\|_F + \frac{\|\mathbf{V}^{(t)}\|_F}{\rho_{t-1}} + \frac{\|\mathbf{V}^{(t-1)}\|_F}{\rho_{t-1}}\right) \leq \frac{C_A}{2}. \tag{16}$$

It then follows from $\mathbf{V}$-update rule and triangle inequality that

$$
\begin{aligned}
\|\mathbf{S}^{(t+1)} - \mathbf{S}^{(t)}\|_F &\leq \|\mathbf{S}^{(t+1)} - \mathbf{D}^{(t+1)}\|_F + \|\mathbf{D}^{(t+1)} - \mathbf{D}^{(t)}\|_F + \|\mathbf{S}^{(t)} - \mathbf{D}^{(t)}\|_F \\
&\leq \frac{\|\mathbf{V}^{(t+1)}\|_F + \|\mathbf{V}^{(t)}\|_F}{\rho_t} + \|\mathbf{D}^{(t+1)} - \mathbf{D}^{(t)}\|_F + \frac{\|\mathbf{V}^{(t)}\|_F + \|\mathbf{V}^{(t-1)}\|_F}{\rho_{t-1}} \\
&\leq \frac{3C_A}{\rho_{t-1}}.
\end{aligned}
\tag{17}
$$

According to **L**-update rule, we have

$$
\begin{aligned}
\|\mathbf{L}^{(t+1)} - \mathbf{L}^{(t)}\|_F &= \left\| \mathbf{H}^{-1/2} P_r(\mathbf{H}^{1/2}(\widehat{\mathbf{W}} - \mathbf{S}^{(t+1)})) - \mathbf{H}^{-1/2} P_r(\mathbf{H}^{1/2}(\widehat{\mathbf{W}} - \mathbf{S}^{(t)})) \right\|_F \\
&\leq \|\mathbf{H}^{-1/2}\|_2 \left\| P_r(\mathbf{H}^{1/2}(\widehat{\mathbf{W}} - \mathbf{S}^{(t+1)})) - P_r(\mathbf{H}^{1/2}(\widehat{\mathbf{W}} - \mathbf{S}^{(t)})) \right\|_F \\
&\leq C_F \|\mathbf{H}^{1/2}\|_2 \|\mathbf{S}^{(t+1)} - \mathbf{S}^{(t)}\|_F \\
&\leq C_F^2 \|\mathbf{S}^{(t+1)} - \mathbf{S}^{(t)}\|_F \\
&\leq \frac{3 C_F^2 C_A}{\rho_{t-1}}.
\end{aligned}
\tag{18}
$$

Therefore, with constant $C = 3 C_F^2 C_A$, we obtain

$$
\max\{\|\mathbf{S}^{(t+1)} - \mathbf{S}^{(t)}\|_F, \|\mathbf{L}^{(t+1)} - \mathbf{L}^{(t)}\|_F\} \leq \frac{C}{\rho_{t-1}}.
\tag{19}
$$

Since $\sum_{s=0}^{\infty} 1/\rho_s < \infty$, both $\{\mathbf{S}^{(t)}\}_{t=0}^{\infty}$ and $\{\mathbf{L}^{(t)}\}_{t=0}^{\infty}$ are Cauchy sequences. Therefore, there exist matrices $\bar{\mathbf{S}}$ and $\bar{\mathbf{L}}$ such that $\mathbf{S}^{(t)} \to \bar{\mathbf{S}}$ and $\mathbf{L}^{(t)} \to \bar{\mathbf{L}}$ as $t \to \infty$. Setting $\bar{\mathbf{W}} = \bar{\mathbf{S}} + \bar{\mathbf{L}}$, we conclude that $\mathbf{S}^{(t)} + \mathbf{L}^{(t)} \to \bar{\mathbf{W}}$ as $t \to \infty$. $\qquad\square$

## A.1 Proof of Lemma A.1

*Proof.* The **L**-update rule in (5), together with (9) yields

$$
\begin{aligned}
\|\mathbf{L}^{(t)}\|_F &= \left\| \mathbf{H}^{-1/2} P_r(\mathbf{H}^{1/2}(\widehat{\mathbf{W}} - \mathbf{S}^{(t)})) \right\|_F \\
&\leq \|\mathbf{H}^{-1/2}\|_2 \left\| P_r(\mathbf{H}^{1/2}(\widehat{\mathbf{W}} - \mathbf{S}^{(t)})) \right\|_F \\
&\leq C_F \left\| \mathbf{H}^{1/2}(\widehat{\mathbf{W}} - \mathbf{S}^{(t)}) \right\|_F \\
&\leq C_F \left\| \mathbf{H}^{1/2} \right\|_2 \|\widehat{\mathbf{W}}\|_F + C_F \left\| \mathbf{H}^{1/2} \right\|_2 \left\| \mathbf{S}^{(t)} \right\|_F \\
&\leq C_F^2 \|\widehat{\mathbf{W}}\|_F + C_F^2 \|\mathbf{S}^{(t)}\|_F,
\end{aligned}
\tag{20}
$$

where the second inequality follows from the non-expansiveness of rank-r projection operator $P_r$ in Frobenius norm. It then follows from the **V**-update rule in (5) that

$$
\begin{aligned}
\|\mathbf{L}^{(t)}\|_F &\leq C_F^2 \|\widehat{\mathbf{W}}\|_F + C_F^2 \|\mathbf{S}^{(t)}\|_F \\
&= C_F^2 \|\widehat{\mathbf{W}}\|_F + C_F^2 \left\| \mathbf{D}^{(t)} + \frac{\mathbf{V}^{(t)} - \mathbf{V}^{(t-1)}}{\rho_{t-1}} \right\|_F \\
&\leq C_F^3 \left( 1 + \|\mathbf{D}^{(t)}\|_F + \frac{\|\mathbf{V}^{(t)}\|_F}{\rho_{t-1}} + \frac{\|\mathbf{V}^{(t-1)}\|_F}{\rho_{t-1}} \right).
\end{aligned}
\tag{21}
$$

$\qquad\square$

## A.2  Proof of Lemma A.2

*Proof.* According to the $\mathbf{S}$-update rule in (5), it holds

$$\mathbf{S}^{(t+1)} - \mathbf{D}^{(t)} + \frac{\mathbf{V}^{(t)}}{\rho_t} = (\mathbf{H} + \rho_t\mathbf{I})^{-1}(\mathbf{G} - \mathbf{H}\mathbf{L}^{(t)} - \mathbf{V}^{(t)} + \rho_t\mathbf{D}^{(t)}) - \mathbf{D}^{(t)} + \frac{\mathbf{V}^{(t)}}{\rho_t}$$

$$= \left((\mathbf{H} + \rho_t\mathbf{I})^{-1}\rho_t - \mathbf{I}\right)\mathbf{D}^{(t)} + (\mathbf{H} + \rho_t\mathbf{I})^{-1}(\mathbf{G} - \mathbf{H}\mathbf{L}^{(t)} - \mathbf{V}^{(t)}) + \frac{\mathbf{V}^{(t)}}{\rho_t}$$

$$= -\frac{1}{\rho_t}\left(\mathbf{I} + \frac{\mathbf{H}}{\rho_t}\right)^{-1}\mathbf{H}\mathbf{D}^{(t)} + \frac{1}{\rho_t}\left(\mathbf{I} + \frac{\mathbf{H}}{\rho_t}\right)^{-1}(\mathbf{G} - \mathbf{H}\mathbf{L}^{(t)} - \mathbf{V}^{(t)}) + \frac{\mathbf{V}^{(t)}}{\rho_t}$$

$$= \frac{1}{\rho_t}\left(\mathbf{I} + \frac{\mathbf{H}}{\rho_t}\right)^{-1}(\mathbf{G} - \mathbf{H}\mathbf{L}^{(t)} - \mathbf{H}\mathbf{D}^{(t)}) + \frac{1}{\rho_t}\left[\mathbf{I} - \left(\mathbf{I} + \frac{\mathbf{H}}{\rho_t}\right)^{-1}\right]\mathbf{V}^{(t)}$$

$$= \frac{1}{\rho_t}\left(\mathbf{I} + \frac{\mathbf{H}}{\rho_t}\right)^{-1}\left(\mathbf{G} - \mathbf{H}\mathbf{L}^{(t)} - \mathbf{H}\mathbf{D}^{(t)} + \frac{\mathbf{H}\mathbf{V}^{(t)}}{\rho_t}\right) \tag{22}$$

Therefore, we obtain

$$\left\|\mathbf{S}^{(t+1)} - \mathbf{D}^{(t)} + \frac{\mathbf{V}^{(t)}}{\rho_t}\right\|_F \leq \frac{1}{\rho_t}\left\|\left(\mathbf{I} + \frac{\mathbf{H}}{\rho_t}\right)^{-1}\right\|_2\left\|\mathbf{G} - \mathbf{H}\mathbf{L}^{(t)} - \mathbf{H}\mathbf{D}^{(t)} + \frac{\mathbf{H}\mathbf{V}^{(t)}}{\rho_t}\right\|_F$$

$$\leq \frac{1}{\rho_t}\left\|\mathbf{G} - \mathbf{H}\mathbf{L}^{(t)} - \mathbf{H}\mathbf{D}^{(t)} + \frac{\mathbf{H}\mathbf{V}^{(t)}}{\rho_t}\right\|_F \tag{23}$$

$$\leq \frac{1}{\rho_t}\left(\|\mathbf{G} - \mathbf{H}\mathbf{L}^{(t)} - \mathbf{H}\mathbf{D}^{(t)}\|_F + \frac{\|\mathbf{H}\mathbf{V}^{(t)}\|_F}{\rho_t}\right).$$

Denote $\tilde{\mathcal{I}} := \{(i,j) \in [N_{in}] \times [N_{out}] \mid \mathbf{D}_{ij}^{(t)} = 0\}$. It follows from the $\mathbf{D}$-update rule and the definition of the projection operator that

$$\left\|\mathbf{D}^{(t+1)} - \mathbf{S}^{(t+1)} - \frac{\mathbf{V}^{(t)}}{\rho_t}\right\|_F^2 = \min_{\substack{\mathcal{I} \subseteq [N_{in}] \times [N_{out}] \\ |\mathcal{I}| = N_{in}N_{out} - k}} \sum_{(i,j) \in \mathcal{I}}\left(\mathbf{S}^{(t+1)} + \frac{\mathbf{V}^{(t)}}{\rho_t}\right)_{i,j}^2$$

$$\leq \sum_{(i,j)\in\tilde{\mathcal{I}}}\left(\mathbf{S}^{(t+1)} + \frac{\mathbf{V}^{(t)}}{\rho_t}\right)_{i,j}^2 = \sum_{(i,j)\in\tilde{\mathcal{I}}}\left(\mathbf{S}^{(t+1)} - \mathbf{D}^{(t)} + \frac{\mathbf{V}^{(t)}}{\rho_t}\right)_{i,j}^2 \tag{24}$$

$$\leq \left\|\mathbf{S}^{(t+1)} - \mathbf{D}^{(t)} + \frac{\mathbf{V}^{(t)}}{\rho_t}\right\|_F^2$$

Together with (23), we get

$$\left\|\mathbf{D}^{(t+1)} - \mathbf{S}^{(t+1)} - \frac{\mathbf{V}^{(t)}}{\rho_t}\right\|_F \leq \frac{1}{\rho_t}\left(\|\mathbf{G} - \mathbf{H}\mathbf{L}^{(t)} - \mathbf{H}\mathbf{D}^{(t)}\|_F + \frac{\|\mathbf{H}\mathbf{V}^{(t)}\|_F}{\rho_t}\right). \tag{25}$$

It then follows from the $\mathbf{V}$-update rule that

$$\frac{\|\mathbf{V}^{(t+1)}\|_F}{\rho_t} = \left\|\mathbf{D}^{(t+1)} - \mathbf{S}^{(t+1)} - \frac{\mathbf{V}^{(t)}}{\rho_t}\right\|_F \leq \frac{1}{\rho_t}\left(\|\mathbf{G} - \mathbf{H}\mathbf{L}^{(t)} - \mathbf{H}\mathbf{D}^{(t)}\|_F + \frac{\|\mathbf{H}\mathbf{V}^{(t)}\|_F}{\rho_t}\right) \tag{26}$$

According to Lemma A.1 and the monotonicity of $\{\rho_t\}_{t=0}^{\infty}$, it holds

$$\|\mathbf{G} - \mathbf{H}\mathbf{L}^{(t)} - \mathbf{H}\mathbf{D}^{(t)}\|_F + \frac{\|\mathbf{H}\mathbf{V}^{(t)}\|_F}{\rho_t} \leq \|\mathbf{G}\|_F + \|\mathbf{H}\|_2\|\mathbf{L}^{(t)}\|_F + \|\mathbf{H}\|_2\|\mathbf{D}^{(t)}\|_F + \frac{\|\mathbf{H}\|_2\|\mathbf{V}^{(t)}\|_F}{\rho_t}$$

$$\leq C_F\left(1 + \|\mathbf{D}^{(t)}\|_F + \frac{\|\mathbf{V}^{(t)}\|_F}{\rho_{t-1}}\right) + C_F\|\mathbf{L}^{(t)}\|_F$$

$$\leq (C_F + C_F^4)\left(1 + \|\mathbf{D}^{(t)}\|_F + \frac{\|\mathbf{V}^{(t)}\|_F}{\rho_{t-1}} + \frac{\|\mathbf{V}^{(t-1)}\|_F}{\rho_{t-1}}\right). \tag{27}$$

Together with inequality (26), this establishes the first inequality of the lemma. Furthermore, by summing up (23) and (25) and applying the triangle inequality, we verify the second inequality. $\square$

## A.3 Proof of Lemma A.3

*Proof.* It follows from Lemma A.2 that

$$\frac{\|\mathbf{V}^{(t+1)}\|_F}{\rho_t} \leq \frac{C_F + C_F^4}{\rho_t}\left(1 + \|\mathbf{D}^{(t)}\|_F + \frac{\|\mathbf{V}^{(t)}\|_F}{\rho_{t-1}} + \frac{\|\mathbf{V}^{(t-1)}\|_F}{\rho_{t-1}}\right) \tag{28}$$

and

$$\begin{aligned}
\|\mathbf{D}^{(t+1)}\|_F &\leq \|\mathbf{D}^{(t)}\|_F + \|\mathbf{D}^{(t+1)} - \mathbf{D}^{(t)}\|_F \\
&\leq \|\mathbf{D}^{(t)}\|_F + \frac{2C_F + 2C_F^4}{\rho_t}\left(1 + \|\mathbf{D}^{(t)}\|_F + \frac{\|\mathbf{V}^{(t)}\|_F}{\rho_{t-1}} + \frac{\|\mathbf{V}^{(t-1)}\|_F}{\rho_{t-1}}\right).
\end{aligned} \tag{29}$$

Summing up these two inequalities yields

$$\begin{aligned}
\|\mathbf{D}^{(t+1)}\|_F + \frac{\|\mathbf{V}^{(t+1)}\|_F}{\rho_t} + \frac{\|\mathbf{V}^{(t)}\|_F}{\rho_t} &\leq \|\mathbf{D}^{(t+1)}\|_F + \frac{\|\mathbf{V}^{(t+1)}\|_F}{\rho_t} + \frac{\|\mathbf{V}^{(t)}\|_F}{\rho_{t-1}} \\
&\leq \frac{3C_F + 3C_F^4}{\rho_t}\left(1 + \|\mathbf{D}^{(t)}\|_F + \frac{\|\mathbf{V}^{(t)}\|_F}{\rho_{t-1}} + \frac{\|\mathbf{V}^{(t-1)}\|_F}{\rho_{t-1}}\right) + \|\mathbf{D}^{(t)}\|_F + \frac{\|\mathbf{V}^{(t)}\|_F}{\rho_{t-1}} \\
&\leq \left(1 + \frac{3C_F + 3C_F^4}{\rho_{t-1}}\right)\left(\|\mathbf{D}^{(t)}\|_F + \frac{\|\mathbf{V}^{(t)}\|_F}{\rho_{t-1}} + \frac{\|\mathbf{V}^{(t-1)}\|_F}{\rho_{t-1}}\right) + \frac{3C_F + 3C_F^4}{\rho_{t-1}},
\end{aligned} \tag{30}$$

Denote $a_t := \|\mathbf{D}^{(t)}\|_F + \|\mathbf{V}^{(t)}\|_F/\rho_{t-1} + \|\mathbf{V}^{(t-1)}\|_F/\rho_{t-1}$, then the above inequality can be rewritten as

$$a_{t+1} \leq \left(1 + \frac{3C_F + 3C_F^4}{\rho_{t-1}}\right)a_t + \frac{3C_F + 3C_F^4}{\rho_{t-1}} \tag{31}$$

Therefore,

$$\begin{aligned}
\frac{a_{t+1}}{\prod_{s=1}^{t}(1 + 3(C_F + C_F^4)/\rho_{s-1})} &\leq \frac{a_t}{\prod_{s=1}^{t-1}(1 + 3(C_F + C_F^4)/\rho_{s-1})} + \frac{3(C_F + C_F^4)}{\rho_{t-1}\prod_{s=0}^{t}(1 + 3(C_F + C_F^4)/\rho_{s-1})} \\
&\leq \frac{a_t}{\prod_{s=1}^{t-1}(1 + 3(C_F + C_F^4)/\rho_{s-1})} + \frac{3(C_F + C_F^4)}{\rho_{t-1}}
\end{aligned} \tag{32}$$

It then follows from telescoping that

$$\frac{a_t}{\prod_{s=1}^{t-1}(1 + 3(C_F + C_F^4)/\rho_{s-1})} \leq a_1 + \sum_{s=1}^{t-1}\frac{3(C_F + C_F^4)}{\rho_{s-1}} \tag{33}$$

Note that

$$\prod_{s=1}^{t-1}(1 + 3(C_F + C_F^4)/\rho_{s-1}) \leq \exp\left(3(C_F + C_F^4)\sum_{s=1}^{t-1}\frac{1}{\rho_{s-1}}\right), \tag{34}$$

recalling the definition of $a_t$ completes the proof. $\qquad\square$

# B Additional Experimental Details

**Computing environments** All experiments were conducted on a computing cluster. Unless otherwise specified, we utilized an Intel Xeon Gold 6248 machine with 16 CPU cores and a single NVIDIA L40 48GB / A100 80GB / H100 80GB GPU. When runtime compression results are reported, all experiments have been run on the same node (including GPU) configuration. All language models and pruning methods were implemented using the PyTorch library Paszke et al. [2017].

**Implementation Details of** `3BASiL` We use $\mathbf{H}' = \mathbf{H} + 0.005\text{diag}(\mathbf{X}^\top\mathbf{X}) + 0.005\,\text{Tr}(\mathbf{X}^\top\mathbf{X})\mathbf{I}$.

In practice, we employ an iteration-dependent penalty parameter $\rho_t$, giving the following updates at iteration $t$:

$$\mathbf{S}^{(t+1)} = (\mathbf{H} + \rho_t\mathbf{I})^{-1}\left(\mathbf{H}(\widehat{\mathbf{W}} - \mathbf{L}^{(t)}) - \mathbf{V}^{(t)} + \rho_t\mathbf{D}^{(t)}\right) \quad \mathbf{L}^{(t+1)} = \mathbf{H}^{-1/2}P_r(\mathbf{H}^{1/2}(\widehat{\mathbf{W}} - \mathbf{S}^{(t+1)}))$$
$$\mathbf{D}^{(t+1)} = P_{\mathcal{S}}(\mathbf{S}^{(t+1)} + \mathbf{V}^{(t)}/\rho_t) \qquad\qquad \mathbf{V}^{(t+1)} = \mathbf{V}^{(t)} + \rho_t(\mathbf{S}^{(t+1)} - \mathbf{D}^{(t+1)}). \tag{35}$$

The initial $\rho_0 = 0.1$. The $\rho$-update for ADMM depends on the support change similar to what was proposed by Meng et al. [2024a]. The $(\mathbf{S} + \mathbf{LR})$ decomposition is more "sensitive" to increasing $\rho$ aggressively compared to pure pruning in the works of Meng et al. [2024a]. We use the following $\rho$ update rules. We update $\rho$ every 10 iteration based on a step function that depends on the current value of $\rho_t$ and $s_t := |\,\text{Supp}\left(\mathbf{D}^{(t)}\right)\Delta\,\text{Supp}\left(\mathbf{D}^{(t-10)}\right)|$, which represents the number of elements in the symmetric difference between $\text{Supp}\left(\mathbf{D}^{(t)}\right)$ and $\text{Supp}\left(\mathbf{D}^{(t-10)}\right)$. Specifically, we set

$$\rho_{t+1} = \begin{cases} 1.1\rho_t & \text{if } s_t \geq 0.1k, \\ 1.05\rho_t & \text{if } s_t \geq 0.005k, \\ 1.02\rho_t & \text{if } s_t \geq 0.5. \end{cases} \tag{36}$$

It is worth noting that the algorithm can converge significantly faster if we set these parameters to the ones proposed by Meng et al. [2024a] (ADMM for pruning) but the solution quality can be slightly compromised.

**Implementation Details of transformer matching** (`TM`) Given a transformer $T_i$ with input activations $\mathbf{X_i}$ (obtained from the outputs of the previously compressed transformer block $T_{i-1}$, we start by creating a copy of $T_i$, termed $T_i^{\mathbf{ori}}$. We then compress $T_i$ layers using an $(\mathbf{S} + \mathbf{LR})$ method. We now replace dense layers with LoRA layers that contain new linear sparse layers and low-rank components $\mathbf{A}, \mathbf{B}$. We set all parameters in transformer block $T_i$ to be trainable and minimize using Adam the loss $\|T_i(\mathbf{X_i}) - T_i^{(ori)}(\mathbf{X_i})\|_F^2$. The input activations fed into subsequent transformer blocks are $T_i^{(TM)}(\mathbf{X_i})$, where $T_i^{(TM)}$ is the transformer block after $(\mathbf{S} + \mathbf{LR})$ decomposition and TM refinement steps.

For `TM` step, we employ the Adam optimizer with PyTorch's default hyperparameters. We use 20 epochs (on the 128 calibration data points selected for compression). The batch size used is 8. The learning rate is $2e^{-5}$ using a Cosine Annealing Scheduler with $\eta_{\min} = 4e^{-6}$.

**Baseline Implementation Details** Below are the implementation specifications for:

- **OATS:** We adopt the official implementation from Zhang and Papyan [2025] (accessible via GitHub) and apply the default hyperparameters and 80 alternating minimization steps.
- **HASSLE-free-SparseGPT:** We adopt the official implementation from Makni et al. [2025] (accessible via GitHub) and provide an improved implementation that uses the closed-form solution Equation (4) for the low-rank fitting step. We apply the default hyperparameters and 80 alternating minimization steps.
- **HASSLE-free-ALPS:** We adopt the official implementation from Makni et al. [2025] (accessible via GitHub) and provide an improved implementation that uses the closed-form solution Equation (4) for the low-rank fitting step. We apply the default hyperparameters and 80 alternating minimization steps.

In Table 3, we use the values reported in Makni et al. [2025]. For all other reported values, instead of minimizing $\|\mathbf{X}(\mathbf{W} - \mathbf{M})\|_F$ s.t. $\text{rank}(\mathbf{M}) \leq r$ by reparameterizing $\mathbf{M} = \mathbf{U}\mathbf{V}^\top$ and optimizing

| Model | Algorithm | Perplexity (↓) | | | Zero-shot (↑) | | | | | | | | |
|---|---|---|---|---|---|---|---|---|---|---|---|---|---|
| | | C4 | WT2 | PTB | PIQA | HS | ARC-E | ARC-C | WG | RTE | OQA | BoolQ | Avg |
| Llama3-8B | Hf-SparseGPT-original | 18.06 | 12.66 | 18.66 | 74.86 | 64.77 | 63.85 | 37.37 | 69.22 | 56.68 | 36.40 | 76.12 | 59.91 |
| | Hf-SparseGPT-ours | 17.77 | 12.38 | 18.71 | 74.81 | 65.04 | 66.16 | 38.57 | 70.09 | 54.87 | 38.40 | 77.71 | 60.71 |
| | EoRA-SparseGPT | 21.89 | 15.69 | 23.91 | 72.25 | 58.87 | 57.70 | 34.56 | 66.30 | 54.87 | 33.80 | 73.58 | 56.49 |
| | Hf-ALPS-original | 16.76 | 11.83 | 17.76 | 75.08 | 66.37 | 63.64 | 37.54 | 69.69 | 64.62 | 37.20 | 77.89 | 61.50 |
| | Hf-ALPS-ours | 16.15 | 11.38 | 16.71 | 75.19 | 67.10 | 64.44 | 38.91 | 69.53 | 59.93 | 39.40 | 78.38 | 61.61 |
| | EoRA-ALPS | 18.69 | 13.61 | 20.55 | 73.99 | 62.21 | 61.07 | 37.20 | 68.59 | 57.40 | 36.00 | 74.16 | 58.83 |

Table 6: Comparison of (original paper), our reproduced results (with improved implementation) and EoRA (reduces to HASSLE-free with alternating minimization steps set to 1) for Llama3-8B under the configuration (2:4 + 64). Perplexity (lower is better), Zero-shot accuracy (higher is better).

with gradient-descent on $\mathbf{U}$ and $\mathbf{V}$ as proposed by the authors, we use our improved implementation of HASSLE-free with closed-form solution Equation (4). This results in significant speedup improvements. A slight improvement in LLM evaluation benchmarks is also sometimes observed using the improved implementation. This is expected because gradient-descent on $\mathbf{U}$ and $\mathbf{V}$ **approximately** solves the reduced-rank regression problem, whereas the closed-form solution is an optimal solution.

Table 6 shows an extract of the differences between the implementation of HASSLE-free proposed in Makni et al. [2025] and ours (using closed-form solution for low-rank update). Moreover, the original paper reports a compression runtime (of a Llama3-8B under a 2:4+64LR configuration) of 20.13 hours using a single A100 80GB GPU, whereas we report a compression runtime (for the same setup) of 15.71 hours in Figure 2 (using a single A100 80GB GPU) thanks to the efficiency of the closed-form solution. It is worth noting that `3BASiL` and `3BASiL-TM` are still over 7 times and 3 times, respectively, faster than HASSLE-free-ALPS, even when using the improved implementation for HASSLE-free.

**LoRA Finetuning Details**   We follow a similar LoRA fine-tuning pipeline to the one introduced in Guo et al. [2024]. For LoRA fine-tuning use a learning rate of 2e-5 and a batch size of size 64 per step. The block size used is $1024$ tokens per batch. The effective batch size is obtained by using a physical batch size of 2 on GPU with 32 gradient accumulation steps before each weight update. Training is conducted on 10% of the first shard of the C4 training dataset, which contains over 15 million tokens. We employ the Adam optimizer with PyTorch's default hyperparameters. A cosine learning rate scheduler is used, with a warm-up ratio of 0.03 and no weight decay applied.

**Layer-wise reconstruction error of `3BASiL`**   Figure 5a and Figure 5b show the objective of Equation (1) attained by `3BASiL` and other $(\mathbf{S} + \mathbf{LR})$ methods for the first transformer block of a 2:4+64LR decomposition of a Llama3-8B model for Attention and MLP layers, respectively.

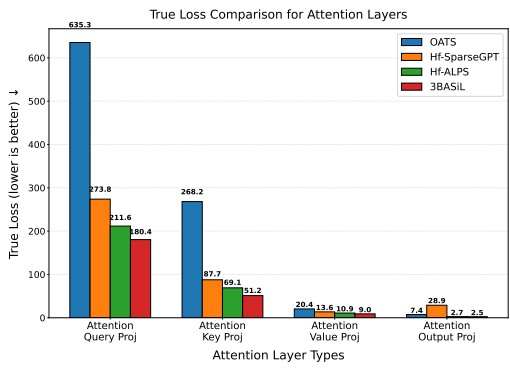

(a) True loss for attention layers (linear scale).

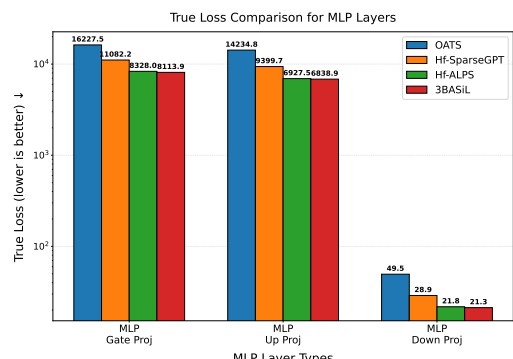

(b) True loss for MLP layers (log scale).

Figure 5: Comparison of true loss values introduced in Equation (1) across different $(\mathbf{S} + \mathbf{LR})$ methods. Lower values indicate better optimization quality. `3BASiL` consistently outperforms other methods, particularly for attention layers.

## C  Additional Experimental Results

We provide additional performance results considered in Section 4. We compare different $(\mathbf{S}+\mathbf{LR})$ algorithms and their TM-enhanced versions (apply TM as an add-on to the decomposition algorithm). In that case, we add the suffix `-TM` to the algorithm. We mark algorithms with TM in gray. We also study the results of the $(\mathbf{S}+\mathbf{LR})$ decomposition after LoRA fine-tuning as described in Appendix B. In that case, we add the prefix `LFT-` to the algorithm.

Example: `LFT-OATS-TM` denotes the results of $(\mathbf{S}+\mathbf{LR})$ decomposition after (i) using OATS to obtain sparse and low-rank components, (ii) refine these decomposed components with TM and (iii) LoRA fine-tunes the model by using the low-rank components from the $(\mathbf{S}+\mathbf{LR})$ decomposition as a smart initialization.

| Method | Config | C4↓ | WT2↓ | PTB↓ | PIQA↑ | HS↑ | ARC-E↑ | ARC-C↑ | WG↑ | RTE↑ | OQA↑ | BoolQ↑ | Avg |
|---|---|---|---|---|---|---|---|---|---|---|---|---|---|
| OATS | | 640.86 | 605.20 | 779.86 | 52.01 | 27.67 | 28.66 | **23.12** | 49.96 | 52.71 | 25.00 | 37.83 | 37.12 |
| OATS-TM | | 116.29 | 99.92 | 126.06 | 55.55 | 28.78 | 31.44 | 20.56 | 51.38 | 52.71 | 26.80 | 46.79 | 39.25 |
| Hassle-free-SparseGPT | | 162.45 | 134.21 | 170.12 | 54.19 | 28.28 | 31.40 | 22.01 | 49.17 | 52.71 | 26.80 | 57.25 | 40.23 |
| Hassle-free-SparseGPT-TM | 2:8+64LR | 74.50 | 67.59 | 88.05 | 57.24 | 30.03 | 33.12 | 21.33 | 52.33 | 52.71 | 26.20 | 60.34 | 41.66 |
| Hassle-free-ALPS | | 107.14 | 94.71 | 124.17 | 55.39 | 29.98 | 32.07 | 20.82 | 52.72 | **53.07** | **27.20** | 49.27 | 40.06 |
| Hassle-free-ALPS-TM | | 58.30 | 52.30 | 73.97 | **58.81** | 32.21 | 34.93 | 21.59 | 52.80 | 52.71 | 26.20 | **62.20** | 42.68 |
| 3BASiL | | 97.50 | 86.59 | 100.35 | 56.91 | 30.49 | 32.37 | 21.08 | **53.75** | **53.07** | 24.40 | 61.74 | 41.73 |
| 3BASiL-TM | | **55.24** | **49.74** | **69.49** | **58.81** | **32.80** | **35.14** | 22.78 | 53.04 | **53.07** | 26.60 | 62.14 | **43.05** |
| OATS | | 125.91 | 92.13 | 115.80 | 57.13 | 32.14 | 35.98 | 22.61 | 51.46 | 51.99 | 26.80 | 62.51 | 42.58 |
| OATS-TM | | 36.32 | 27.69 | 41.98 | 63.17 | 38.47 | 43.69 | 23.21 | 52.88 | 52.71 | 29.80 | 62.08 | 45.75 |
| Hassle-free-SparseGPT | | 43.50 | 34.18 | 51.16 | 61.86 | 38.24 | 41.29 | 24.66 | 53.75 | 52.71 | 29.20 | 62.08 | 45.47 |
| Hassle-free-SparseGPT-TM | 3:8+64LR | 30.48 | 24.18 | 37.28 | 65.18 | 41.40 | 45.41 | 25.68 | 55.96 | 52.71 | 30.40 | 62.26 | 47.38 |
| Hassle-free-ALPS | | 37.80 | 29.00 | 43.60 | 64.04 | 41.47 | 42.34 | 25.85 | 54.22 | **54.51** | 30.40 | 62.02 | 46.86 |
| Hassle-free-ALPS-TM | | 27.34 | 21.42 | 34.23 | 66.38 | 44.48 | 45.62 | 25.51 | 55.88 | 53.43 | 30.40 | 62.23 | 47.99 |
| 3BASiL | | 34.81 | 26.96 | 41.55 | 64.91 | 42.34 | 45.24 | **27.47** | 56.12 | 53.07 | 31.20 | **62.60** | 47.87 |
| 3BASiL-TM | | **26.26** | **20.75** | **32.09** | **66.43** | **45.47** | **47.47** | 27.05 | **57.77** | 52.71 | **32.00** | 62.20 | **48.89** |
| OATS | | 28.06 | 19.69 | 32.90 | 67.30 | 49.07 | 48.48 | 27.65 | 56.20 | 52.71 | 30.20 | 62.45 | 49.26 |
| OATS-TM | | 20.65 | 14.59 | 24.50 | 69.37 | 51.37 | 55.05 | 31.40 | 57.22 | 54.51 | 30.60 | 62.81 | 51.54 |
| Hassle-free-SparseGPT | | 22.24 | 15.90 | 27.35 | 70.51 | 52.08 | 50.76 | 29.44 | 57.38 | **58.12** | **34.20** | 62.63 | 51.89 |
| Hassle-free-SparseGPT-TM | 4:8+64LR | 19.63 | 13.85 | 24.17 | 70.24 | 53.12 | 55.05 | 31.14 | 56.83 | 54.87 | 33.80 | 63.18 | 52.28 |
| Hassle-free-ALPS | | 20.71 | 14.90 | 24.75 | 69.59 | 53.27 | 53.07 | 29.78 | 57.77 | 53.79 | 33.60 | 63.33 | 51.78 |
| Hassle-free-ALPS-TM | | 19.07 | 13.70 | 23.06 | 71.11 | 54.82 | 55.30 | **31.83** | 58.64 | 52.71 | 31.60 | 62.42 | 52.30 |
| 3BASiL | | 20.04 | 14.26 | 24.27 | 70.62 | 54.55 | **55.72** | 30.72 | **60.06** | 55.23 | 34.00 | 63.06 | 52.99 |
| 3BASiL-TM | | **18.66** | **13.19** | **22.46** | **72.09** | **55.48** | 55.60 | 31.48 | 59.12 | 53.07 | 34.00 | **63.46** | **53.04** |
| OATS | | 41.80 | 28.45 | 45.36 | 63.28 | 41.89 | 47.01 | 26.37 | 53.51 | 51.26 | 28.40 | 63.09 | 46.85 |
| OATS-TM | | 23.89 | 17.04 | 27.99 | 68.23 | 47.76 | 51.01 | 27.73 | 55.64 | 56.32 | 32.60 | 62.32 | 50.20 |
| Hassle-free-SparseGPT | | 27.25 | 19.45 | 32.63 | 67.63 | 47.70 | 45.96 | 26.96 | 55.88 | 52.71 | 30.40 | 62.14 | 48.67 |
| Hassle-free-SparseGPT-TM | 2:4+64LR | 22.17 | 16.41 | 26.67 | 69.10 | 49.90 | 50.29 | 27.99 | 56.59 | **57.04** | 33.40 | 62.42 | 50.84 |
| Hassle-free-ALPS | | 23.90 | 17.66 | 28.96 | 69.15 | 49.62 | 49.66 | 28.16 | 57.77 | 55.23 | 32.00 | 63.06 | 50.58 |
| Hassle-free-ALPS-TM | | 20.93 | 15.35 | 25.15 | **70.46** | 51.14 | 51.09 | 28.24 | 58.33 | **57.04** | **34.60** | **63.55** | **51.81** |
| 3BASiL | | 23.16 | 17.27 | 27.77 | 69.80 | 51.74 | 51.35 | 27.82 | 58.72 | 54.87 | 33.40 | 62.84 | 51.32 |
| 3BASiL-TM | | **20.46** | **15.23** | **24.60** | 70.18 | **52.91** | **52.06** | **30.12** | **58.96** | 52.71 | 33.40 | 62.39 | 51.59 |
| **Llama-3.2-1B Dense** | – | 14.01 | 9.75 | 17.59 | 74.59 | 63.66 | 60.48 | 36.26 | 60.69 | 56.68 | 37.20 | 63.98 | 56.69 |

Table 7: One-shot (N:M Sparse + LR) decomposition performance for Llama-3.2-1B. For Perplexity, (↓) lower values are better. For zero-shot tasks, (↑) higher values are better.

| Method | Config | C4↓ | WT2↓ | PTB↓ | PIQA↑ | HS↑ | ARC-E↑ | ARC-C↑ | WG↑ | RTE↑ | OQA↑ | BoolQ↑ | Avg |
|---|---|---|---|---|---|---|---|---|---|---|---|---|---|
| OATS | | 531.47 | 494.31 | 674.71 | 52.50 | 27.33 | 28.16 | **23.29** | 49.57 | 52.71 | 26.60 | 39.60 | 37.47 |
| OATS-TM | | 100.87 | 87.20 | 120.98 | 56.64 | 29.01 | 30.13 | 20.65 | 50.99 | 52.71 | 26.00 | 62.11 | 41.03 |
| Hassle-free-SparseGPT | | 106.07 | 106.17 | 151.92 | 54.62 | 29.67 | 29.92 | 21.67 | 50.28 | 52.71 | 26.60 | 61.93 | 40.93 |
| Hassle-free-SparseGPT-TM | 2:8+64LR | 61.50 | 56.02 | 90.37 | 58.98 | 32.53 | 33.75 | 22.10 | 51.78 | 52.71 | 26.80 | 62.11 | 42.55 |
| Hassle-free-ALPS | | 69.96 | 65.34 | 108.68 | 57.34 | 32.59 | 33.59 | 20.82 | 50.67 | 52.71 | **27.00** | 62.26 | 42.12 |
| Hassle-free-ALPS-TM | | 46.12 | 44.03 | **61.25** | **61.26** | 36.36 | 36.83 | 23.12 | 52.80 | 52.71 | 25.00 | 62.51 | 43.82 |
| 3BASiL | | 73.00 | 72.26 | 110.10 | 57.29 | 32.62 | 34.01 | 21.42 | 51.14 | 52.71 | 26.80 | 62.20 | 42.27 |
| 3BASiL-TM | | **45.35** | **42.38** | 68.29 | 61.10 | **36.90** | **38.17** | 22.78 | **53.12** | **53.07** | 26.00 | **62.66** | **44.23** |
| OATS | | 65.08 | 47.27 | 81.29 | 61.75 | 37.80 | 42.17 | 23.89 | 52.88 | 52.71 | 27.20 | 62.75 | 45.14 |
| OATS-TM | | 27.09 | 20.94 | 30.21 | 67.68 | 47.26 | 51.26 | 28.41 | 57.46 | 52.71 | 29.20 | 64.65 | 49.83 |
| Hassle-free-SparseGPT | | 34.66 | 26.60 | 39.76 | 65.94 | 46.19 | 47.77 | 26.88 | 58.96 | 53.07 | 29.60 | 65.02 | 49.18 |
| Hassle-free-SparseGPT-TM | 3:8+64LR | 23.69 | 19.54 | 27.45 | 69.70 | 51.64 | 52.78 | 29.35 | 60.22 | 55.96 | 30.60 | 62.72 | 51.62 |
| Hassle-free-ALPS | | 27.94 | 22.77 | 34.59 | 69.15 | 50.18 | 53.32 | 29.01 | **61.48** | 52.71 | 32.00 | 63.58 | 51.43 |
| Hassle-free-ALPS-TM | | 21.52 | 17.80 | 26.42 | 71.00 | 54.45 | **57.37** | 30.80 | 59.75 | **56.32** | **33.40** | 66.02 | **53.64** |
| 3BASiL | | 26.35 | 20.66 | 31.77 | 68.66 | 51.44 | 52.10 | 29.95 | 61.25 | 54.15 | 31.20 | **68.32** | 52.13 |
| 3BASiL-TM | | **20.89** | **17.18** | **25.31** | **71.82** | **55.35** | 55.35 | **32.08** | 60.85 | 54.15 | **33.40** | 65.41 | 53.53 |
| OATS | | 19.25 | 13.40 | 21.67 | 72.47 | 61.10 | 60.82 | 35.15 | 66.30 | 57.40 | 34.00 | 73.24 | 57.56 |
| OATS-TM | | 15.92 | 11.00 | 17.82 | 74.21 | 63.46 | 65.61 | 37.80 | 66.14 | **64.62** | 36.80 | 72.26 | 60.11 |
| Hassle-free-SparseGPT | | 17.09 | 12.30 | 19.19 | 73.83 | 63.01 | 64.23 | 36.52 | 65.59 | 58.12 | 37.80 | 72.08 | 58.90 |
| Hassle-free-SparseGPT-TM | 4:8+64LR | 15.42 | 10.84 | 17.23 | 74.65 | 65.22 | 64.94 | 38.14 | 65.19 | 58.84 | **40.40** | 69.69 | 59.63 |
| Hassle-free-ALPS | | 16.04 | 11.51 | 18.17 | 74.54 | 64.63 | 63.76 | 36.95 | 66.38 | 59.57 | 36.80 | 72.08 | 59.34 |
| Hassle-free-ALPS-TM | | 15.07 | 10.54 | 16.84 | 75.19 | 65.93 | 66.84 | **40.02** | **67.40** | 60.65 | 40.00 | **74.01** | **61.25** |
| 3BASiL | | 15.65 | 10.97 | 17.39 | 75.68 | 65.87 | 66.46 | 39.42 | 67.25 | 59.93 | 38.80 | 73.52 | 60.87 |
| 3BASiL-TM | | **14.89** | **10.29** | **16.52** | **75.79** | **66.46** | **67.05** | 38.82 | 66.06 | 59.93 | 39.20 | 72.32 | 60.70 |
| OATS | | 25.18 | 17.41 | 28.60 | 70.89 | 54.76 | 57.74 | 32.76 | 61.17 | 53.07 | 32.80 | 70.40 | 54.20 |
| OATS-TM | | 18.08 | 12.85 | 20.22 | 72.42 | 59.46 | 62.79 | 35.24 | 62.27 | 58.84 | 34.60 | 70.21 | 56.98 |
| Hassle-free-SparseGPT | | 20.38 | 15.03 | 23.23 | 71.55 | 58.62 | 59.93 | 32.94 | 63.85 | 57.40 | 33.60 | 69.94 | 55.98 |
| Hassle-free-SparseGPT-TM | 2:4+64LR | 17.24 | 12.66 | 19.41 | 73.78 | 61.50 | 61.66 | 34.64 | 63.69 | 58.48 | 37.20 | 68.62 | 57.45 |
| Hassle-free-ALPS | | 18.45 | 13.79 | 20.50 | 73.78 | 60.82 | **63.30** | 35.49 | 64.56 | 57.40 | 35.80 | **72.78** | 57.99 |
| Hassle-free-ALPS-TM | | 16.60 | 12.11 | 18.59 | **74.21** | 62.97 | 63.05 | 37.54 | 66.14 | 58.84 | 36.00 | 70.58 | 58.67 |
| 3BASiL | | 17.89 | 13.12 | 20.10 | 73.34 | 61.99 | 62.50 | 35.07 | **66.46** | **61.73** | **39.60** | 71.80 | **59.06** |
| 3BASiL-TM | | **16.37** | **11.79** | **18.34** | 73.78 | **63.38** | 63.05 | **38.05** | 64.25 | 59.57 | 36.80 | 71.13 | 58.75 |
| **Llama-3.2-3B Dense** | – | 11.33 | 7.81 | 13.53 | 77.48 | 73.61 | 71.63 | 45.99 | 69.85 | 54.51 | 43.00 | 73.39 | 63.68 |

Table 8: One-shot (N:M Sparse + LR) decomposition performance for Llama-3.2-3B. For Perplexity, (↓) lower values are better. For zero-shot tasks, (↑) higher values are better.

| Method | Config | C4↓ | WT2↓ | PTB↓ | PIQA↑ | HS↑ | ARC-E↑ | ARC-C↑ | WG↑ | RTE↑ | OQA↑ | BoolQ↑ | Avg |
|---|---|---|---|---|---|---|---|---|---|---|---|---|---|
| OATS | 2:8+64LR | 424.55 | 431.81 | 590.88 | 51.63 | 28.11 | 27.61 | 23.72 | 49.72 | **52.71** | 27.00 | 38.26 | 37.34 |
| OATS-TM | | 73.42 | 64.21 | 100.59 | 58.43 | 31.23 | 32.20 | 20.22 | 51.38 | **52.71** | 26.00 | 62.29 | 41.81 |
| Hassle-free-SparseGPT | | 88.39 | 96.61 | 109.71 | 54.95 | 30.77 | 31.48 | 20.65 | 50.75 | **52.71** | 26.20 | 61.59 | 41.14 |
| Hassle-free-SparseGPT-TM | | 46.24 | 42.75 | 71.66 | 61.48 | 36.76 | 36.74 | 23.12 | 53.51 | **52.71** | 27.60 | 63.15 | 44.38 |
| Hassle-free-ALPS | | 60.16 | 56.03 | 77.11 | 57.94 | 34.92 | 34.64 | 21.67 | 54.62 | **52.71** | 27.80 | 56.12 | 42.55 |
| Hassle-free-ALPS-TM | | 36.50 | 34.31 | **50.14** | **64.91** | 41.18 | **40.28** | **24.40** | 56.99 | **52.71** | 28.20 | 59.60 | 46.03 |
| 3BASiL | | 56.99 | 53.83 | 72.48 | 59.25 | 35.34 | 35.98 | 21.59 | 54.06 | **52.71** | 27.40 | **64.62** | 43.87 |
| 3BASiL-TM | | **36.16** | **33.51** | 52.87 | 63.60 | **41.47** | 39.81 | 24.32 | **58.41** | **52.71** | 26.80 | 63.67 | **46.35** |
| OATS | 3:8+64LR | 58.88 | 40.76 | 67.35 | 63.71 | 39.48 | 42.68 | 24.32 | 53.91 | 52.71 | 28.40 | 63.98 | 46.15 |
| OATS-TM | | 22.67 | 17.17 | 24.46 | 71.22 | 54.29 | 54.88 | 31.48 | 63.22 | 54.15 | 32.80 | 71.38 | 54.18 |
| Hassle-free-SparseGPT | | 29.32 | 21.46 | 32.06 | 68.66 | 51.99 | 50.97 | 30.38 | 63.85 | 53.07 | 32.00 | 71.31 | 52.78 |
| Hassle-free-SparseGPT-TM | | 19.97 | 15.51 | 22.28 | 72.85 | 58.72 | 58.12 | 33.96 | 65.67 | **61.37** | 32.00 | 74.92 | 57.20 |
| Hassle-free-ALPS | | 23.93 | 18.20 | 26.31 | 70.62 | 56.54 | 54.42 | 30.12 | 64.72 | 55.23 | 32.80 | 71.96 | 54.55 |
| Hassle-free-ALPS-TM | | 18.38 | 14.52 | **20.15** | **74.48** | 61.56 | 59.01 | 33.28 | **67.48** | 57.40 | **35.20** | 75.11 | 57.94 |
| 3BASiL | | 23.07 | 18.03 | 24.84 | 71.06 | 56.96 | 57.70 | 32.59 | 66.69 | 54.51 | 33.00 | 66.70 | 54.90 |
| 3BASiL-TM | | **18.11** | **14.26** | 20.47 | 74.05 | **61.85** | **60.73** | **34.73** | 65.94 | 54.51 | 34.80 | **76.91** | **57.94** |
| OATS | 4:8+64LR | 16.38 | 10.88 | 17.23 | 75.84 | 67.60 | 67.09 | 41.21 | 70.88 | 60.29 | 38.20 | 73.61 | 61.84 |
| OATS-TM | | 13.77 | 9.06 | 14.37 | 76.82 | 70.15 | 70.16 | 43.52 | 70.88 | **65.70** | 40.20 | 77.89 | 64.41 |
| Hassle-free-SparseGPT | | 14.65 | 9.88 | 15.21 | 77.09 | 69.95 | 69.32 | 41.81 | 71.27 | 56.32 | 40.60 | 79.39 | 63.22 |
| Hassle-free-SparseGPT-TM | | 13.40 | 8.90 | 14.11 | 77.58 | 71.42 | **73.23** | 43.60 | 70.40 | 64.98 | 41.40 | 79.39 | 65.25 |
| Hassle-free-ALPS | | 14.04 | 9.44 | 14.45 | 76.82 | 71.19 | 71.04 | 44.45 | **72.77** | 56.68 | 40.20 | 78.13 | 63.91 |
| Hassle-free-ALPS-TM | | 13.21 | 8.71 | 13.85 | **78.56** | 72.54 | 72.81 | 45.73 | 71.43 | 65.34 | 41.40 | 79.88 | 65.96 |
| 3BASiL | | 13.74 | 9.21 | 14.24 | 76.88 | 72.05 | 70.16 | 44.80 | 72.14 | 61.01 | 41.40 | **80.89** | 64.92 |
| 3BASiL-TM | | **13.02** | **8.64** | **13.70** | 78.24 | **72.59** | 73.11 | **47.35** | 72.00 | 63.18 | **42.40** | 80.49 | **66.17** |
| OATS | 2:4+64LR | 21.59 | 14.76 | 23.41 | 72.74 | 60.70 | 60.86 | 34.81 | 65.51 | 57.76 | 35.20 | 68.32 | 56.99 |
| OATS-TM | | 15.49 | 10.61 | 16.11 | 76.01 | 65.66 | 67.00 | 40.61 | 68.59 | 56.68 | 36.60 | 75.69 | 60.86 |
| Hassle-free-SparseGPT | | 17.77 | 12.38 | 18.71 | 74.81 | 65.04 | 66.16 | 38.57 | 67.09 | 54.87 | 34.80 | 77.71 | 60.71 |
| Hassle-free-SparseGPT-TM | | 14.95 | 10.28 | 15.97 | 76.88 | 68.18 | 67.21 | **41.81** | 69.46 | 64.98 | 38.20 | 78.81 | 63.19 |
| Hassle-free-ALPS | | 16.15 | 11.38 | 16.71 | 75.19 | 67.10 | 64.44 | 38.91 | 69.59 | 59.93 | 39.40 | 78.38 | 61.61 |
| Hassle-free-ALPS-TM | | 14.45 | 10.00 | 15.23 | 77.09 | 69.32 | **67.26** | 40.10 | 70.17 | 60.65 | 38.80 | 75.75 | 62.39 |
| 3BASiL | | 15.76 | 11.23 | 16.25 | 76.50 | 67.61 | 67.21 | 40.10 | 70.24 | 64.26 | 38.20 | 78.29 | 62.80 |
| 3BASiL-TM | | **14.34** | **9.78** | **14.88** | **77.48** | **69.58** | 67.21 | 40.53 | **71.27** | 61.37 | **39.80** | **79.51** | **63.34** |
| **Meta-Llama-3-8B Dense** | – | 9.44 | 6.14 | 11.18 | 80.79 | 79.17 | 77.69 | 53.33 | 72.85 | 69.68 | 45.00 | 81.44 | 69.99 |

Table 9: One-shot (N:M Sparse + LR) decomposition performance for Meta-Llama-3-8B. For Perplexity, (↓) lower values are better. For zero-shot tasks, (↑) higher values are better.

| Method | Config | C4↓ | WT2↓ | PTB↓ | PIQA↑ | HS↑ | ARC-E↑ | ARC-C↑ | WG↑ | RTE↑ | OQA↑ | BoolQ↑ | Avg |
|---|---|---|---|---|---|---|---|---|---|---|---|---|---|
| LFT-OATS | 2:8+64LR | 71.41 | 63.58 | 95.05 | 56.58 | 29.22 | 30.72 | 22.10 | 53.83 | 52.71 | 23.00 | 61.10 | 41.16 |
| LFT-OATS-TM | | 53.01 | 47.68 | 68.98 | 59.09 | 31.07 | 36.15 | 21.84 | 52.49 | 52.71 | 24.60 | 62.14 | 42.51 |
| LFT-Hassle-free-SparseGPT | | 51.19 | 47.09 | 63.87 | 59.03 | 31.16 | 35.14 | 22.18 | 50.83 | 52.71 | 26.60 | 60.70 | 42.29 |
| LFT-Hassle-free-SparseGPT-TM | | 42.85 | 39.46 | 55.53 | **60.55** | 33.37 | 35.27 | 22.18 | 50.36 | 52.71 | 28.00 | 62.14 | 43.07 |
| LFT-Hassle-free-ALPS | | 44.15 | 41.02 | 57.45 | 59.36 | 33.22 | 34.68 | 22.18 | 51.93 | 52.71 | 26.60 | 60.95 | 42.70 |
| LFT-Hassle-free-ALPS-TM | | 37.54 | 35.50 | 50.15 | 60.28 | 35.80 | 35.80 | 23.63 | 55.09 | 52.71 | 27.20 | 62.14 | 44.29 |
| LFT-3BASiL | | 36.09 | **33.42** | **45.13** | **60.55** | 35.52 | **37.71** | 23.21 | 53.12 | **53.07** | 27.00 | 62.02 | 44.02 |
| LFT-3BASiL-TM | | **36.03** | 34.13 | 47.11 | 60.07 | **36.16** | 37.29 | **24.66** | 55.41 | **53.07** | 29.40 | 62.32 | **44.80** |
| LFT-OATS | 3:8+64LR | 32.20 | 25.47 | 41.74 | 62.35 | 39.71 | 44.36 | 25.68 | 53.79 | 52.71 | 29.20 | 62.14 | 46.20 |
| LFT-OATS-TM | | 27.05 | 21.14 | 34.83 | 65.51 | 43.37 | 46.25 | 26.62 | 54.93 | 52.71 | 29.60 | 61.99 | 47.62 |
| LFT-Hassle-free-SparseGPT | | 27.17 | 21.96 | 34.55 | 65.72 | 44.09 | 44.99 | 26.79 | 55.17 | 52.71 | 30.60 | 61.59 | 47.71 |
| LFT-Hassle-free-SparseGPT-TM | | 24.74 | 20.38 | 32.36 | 67.03 | 45.71 | 46.62 | 26.62 | 55.01 | 52.71 | 30.20 | 62.02 | 48.46 |
| LFT-Hassle-free-ALPS | | 25.20 | 20.03 | 32.26 | 65.61 | 46.19 | 45.12 | 28.16 | 54.85 | **55.96** | 32.00 | 62.63 | 48.81 |
| LFT-Hassle-free-ALPS-TM | | 23.42 | 18.81 | 30.32 | **68.23** | 47.85 | 46.59 | 27.22 | 55.25 | 54.51 | 31.20 | 62.23 | 49.14 |
| LFT-3BASiL | | 22.97 | **18.23** | **29.74** | 67.36 | 48.30 | 48.36 | **29.52** | 55.72 | 54.87 | 31.80 | 62.72 | 49.83 |
| LFT-3BASiL-TM | | **22.73** | 18.29 | 29.94 | 68.01 | **49.22** | **48.86** | 29.35 | **56.83** | 52.71 | 33.20 | 63.21 | **50.17** |
| LFT-OATS | 4:8+64LR | 28.06 | 19.69 | 32.90 | 67.30 | 49.07 | 48.48 | 27.65 | 52.33 | 52.71 | 30.20 | 62.45 | 49.26 |
| LFT-OATS-TM | | 18.95 | 13.77 | 23.65 | 70.73 | 54.25 | 56.36 | 32.51 | 58.80 | 54.87 | 33.60 | 62.72 | 52.98 |
| LFT-Hassle-free-SparseGPT | | 19.38 | 14.15 | 24.74 | 71.71 | 53.78 | 53.58 | 30.80 | 56.35 | 57.76 | 33.20 | 60.67 | 52.23 |
| LFT-Hassle-free-SparseGPT-TM | | 18.45 | 13.35 | 23.60 | 71.55 | 55.21 | 56.02 | **32.94** | 56.43 | 53.07 | 36.00 | 63.43 | 53.08 |
| LFT-Hassle-free-ALPS | | 18.77 | 13.82 | 23.63 | 71.06 | 55.59 | 55.01 | 30.29 | 56.99 | 53.79 | 33.80 | 62.91 | 52.43 |
| LFT-Hassle-free-ALPS-TM | | 18.11 | 13.28 | 22.75 | 71.93 | **56.97** | **56.99** | 32.00 | 56.35 | 54.51 | 33.80 | 62.66 | 53.54 |
| LFT-3BASiL | | 17.88 | 12.99 | 22.56 | **72.74** | 56.91 | 56.86 | 31.66 | 60.62 | 59.57 | 36.40 | 62.39 | 54.64 |
| LFT-3BASiL-TM | | **17.75** | **12.82** | **22.30** | 72.52 | 56.81 | 56.81 | 32.94 | 59.51 | 51.62 | 35.20 | **63.61** | 53.58 |
| LFT-OATS | 2:4+64LR | 23.55 | 17.52 | 29.60 | 67.08 | 48.04 | 49.28 | 27.90 | 55.33 | 54.15 | 31.60 | 62.57 | 49.49 |
| LFT-OATS-TM | | 21.00 | 15.40 | 25.77 | 69.80 | 50.92 | 52.15 | 29.01 | 55.96 | 55.23 | 33.00 | 62.63 | 51.09 |
| LFT-Hassle-free-SparseGPT | | 21.56 | 15.99 | 26.81 | 69.53 | 51.03 | 48.65 | 28.67 | 56.04 | 52.35 | 31.40 | 62.14 | 49.98 |
| LFT-Hassle-free-SparseGPT-TM | | 20.16 | 15.16 | 25.23 | 71.11 | 53.40 | 52.86 | 29.69 | 57.22 | **56.68** | 35.00 | 62.17 | 52.27 |
| LFT-Hassle-free-ALPS | | 20.38 | 15.57 | 25.47 | 71.16 | 52.53 | 53.41 | 29.69 | 56.99 | 54.87 | 31.40 | 62.84 | 51.89 |
| LFT-Hassle-free-ALPS-TM | | 19.42 | 14.51 | 24.31 | 71.16 | 54.01 | 53.16 | 28.92 | 58.01 | 54.15 | 34.20 | 60.43 | 51.75 |
| LFT-3BASiL | | 19.25 | 14.60 | 24.56 | **71.76** | 54.72 | 53.16 | 29.01 | 57.77 | 53.43 | 34.00 | **63.15** | 52.12 |
| LFT-3BASiL-TM | | **19.07** | **14.37** | **23.88** | 71.44 | **55.12** | **54.04** | 29.95 | 57.95 | 53.43 | 33.60 | 62.69 | 52.30 |
| **Llama-3.2-1B Dense** | – | 14.01 | 9.75 | 17.59 | 74.59 | 63.66 | 60.48 | 36.26 | 60.69 | 56.68 | 37.20 | 63.98 | 56.69 |

Table 10: (N:M Sparse + LR) decomposition performance for Llama-3.2-1B after LoRa Fine-Tuning (LFT). For Perplexity, (↓) lower values are better. For zero-shot tasks, (↑) higher values are better.

| Method | Config | C4 ↓ | WT2 ↓ | PTB ↓ | PIQA ↑ | HS ↑ | ARC-E ↑ | ARC-C ↑ | WG ↑ | RTE ↑ | OQA ↑ | BoolQ ↑ | Avg |
|---|---|---|---|---|---|---|---|---|---|---|---|---|---|
| LFT-OATS | | 48.53 | 44.51 | 62.21 | 58.27 | 32.84 | 36.95 | 22.78 | 52.96 | 52.71 | 25.60 | 61.44 | 42.94 |
| LFT-OATS-TM | | 100.87 | 87.20 | 120.98 | 56.64 | 29.01 | 30.13 | 20.65 | 50.99 | 52.71 | 26.00 | 62.11 | 41.03 |
| LFT-Hassle-free-SparseGPT | | 35.91 | 33.44 | 46.64 | 61.48 | 37.33 | 36.70 | 23.46 | 51.54 | 52.35 | 26.17 | 62.17 | 43.90 |
| LFT-Hassle-free-SparseGPT-TM | 2:8+64LR | 31.46 | 30.77 | 42.12 | 62.79 | 40.19 | 38.76 | 23.63 | 54.78 | 53.07 | 29.00 | 58.32 | 45.07 |
| LFT-Hassle-free-ALPS | | 31.43 | 29.50 | 44.39 | 63.38 | 40.65 | 39.73 | 23.46 | 55.72 | 52.71 | 28.40 | 46.06 | 43.76 |
| LFT-Hassle-free-ALPS-TM | | 28.22 | 26.81 | **36.23** | 65.02 | 43.69 | 40.78 | 23.46 | 55.72 | 52.71 | 28.40 | 60.46 | 46.28 |
| LFT-3BASiL | | 30.58 | 28.27 | 41.30 | 63.11 | 41.25 | 41.75 | 24.23 | 55.09 | 52.71 | 28.40 | **63.88** | 46.30 |
| LFT-3BASiL-TM | | **27.48** | **26.18** | 38.35 | **65.34** | **44.37** | **43.18** | **26.54** | **57.46** | 53.43 | **29.20** | 61.04 | **47.57** |
| LFT-OATS | | 21.98 | 16.44 | 27.23 | 69.59 | 52.26 | 52.78 | 29.44 | 57.22 | 59.21 | 31.00 | 64.07 | 51.95 |
| LFT-OATS-TM | | 27.09 | 20.94 | 30.21 | 67.68 | 47.26 | 51.26 | 28.41 | 57.46 | 52.71 | 29.20 | 64.65 | 49.83 |
| LFT-Hassle-free-SparseGPT | | 20.01 | 15.72 | 24.87 | 70.46 | 56.25 | 54.59 | 30.72 | 60.22 | 58.12 | 33.20 | 63.64 | 53.40 |
| LFT-Hassle-free-SparseGPT-TM | 3:8+64LR | 18.73 | 15.18 | 23.25 | 71.00 | 58.49 | 54.21 | 32.51 | 61.72 | 59.21 | 35.40 | 53.00 | 53.19 |
| LFT-Hassle-free-ALPS | | 18.83 | 15.54 | 22.96 | 71.60 | 57.15 | 57.15 | 32.51 | **63.22** | 54.87 | 33.80 | 66.64 | 54.76 |
| LFT-Hassle-free-ALPS-TM | | 18.04 | 14.68 | 23.08 | 72.91 | 60.26 | **60.23** | **34.22** | 61.09 | **62.82** | **36.00** | 58.90 | **55.80** |
| LFT-3BASiL | | 18.40 | 14.61 | 22.90 | 72.20 | 59.70 | 57.83 | 32.17 | 62.67 | 55.23 | 35.20 | **68.47** | 55.43 |
| LFT-3BASiL-TM | | **17.69** | **14.38** | **22.42** | **72.96** | **61.21** | 57.32 | 34.04 | 61.80 | 58.48 | 34.80 | 55.81 | 54.55 |
| LFT-OATS | | 15.50 | 10.77 | 17.94 | 75.63 | 65.47 | 65.24 | 38.65 | 65.98 | 57.40 | 37.80 | 65.69 | 58.98 |
| LFT-OATS-TM | | 15.92 | 11.00 | 17.82 | 74.21 | 63.46 | 65.61 | 37.80 | 64.62 | | 36.80 | 72.26 | 60.11 |
| LFT-Hassle-free-SparseGPT | | 14.97 | 10.62 | 17.48 | 75.24 | 66.87 | 65.87 | 39.25 | 66.22 | 58.12 | 38.60 | 72.11 | 60.29 |
| LFT-Hassle-free-SparseGPT-TM | 4:8+64LR | 14.47 | 10.25 | 16.92 | 75.68 | 67.79 | 66.37 | 39.76 | 65.96 | 55.96 | **42.80** | 67.86 | 60.35 |
| LFT-Hassle-free-ALPS | | 14.60 | 10.33 | 16.73 | 75.79 | 67.42 | 64.27 | 37.63 | 65.82 | 62.82 | 38.20 | 55.78 | 58.47 |
| LFT-Hassle-free-ALPS-TM | | 14.30 | 10.11 | 16.50 | 75.63 | 68.18 | 66.75 | **41.21** | **67.88** | 59.21 | 40.00 | 69.39 | 61.03 |
| LFT-3BASiL | | 14.38 | 10.10 | 16.56 | 76.77 | 68.06 | **67.26** | 40.70 | 67.48 | 59.57 | 39.60 | 68.35 | 60.97 |
| LFT-3BASiL-TM | | **14.15** | **9.89** | **16.29** | **77.26** | **68.44** | 66.20 | 39.59 | 66.69 | 62.82 | 39.80 | 72.29 | **61.64** |
| LFT-OATS | | 25.18 | 12.08 | 20.26 | 73.99 | 61.58 | 62.88 | 36.09 | 63.46 | 61.73 | 36.60 | 62.63 | 57.37 |
| LFT-OATS-TM | | 18.08 | 12.85 | 20.22 | 72.42 | 59.46 | 62.79 | 35.24 | 62.27 | 58.84 | 34.60 | **70.21** | 56.98 |
| LFT-Hassle-free-SparseGPT | | 16.36 | 11.79 | 19.43 | 73.94 | 63.71 | 63.51 | 35.07 | 64.01 | 51.62 | 36.80 | 68.99 | 57.21 |
| LFT-Hassle-free-SparseGPT-TM | 2:4+64LR | 15.65 | 11.37 | 18.41 | **75.14** | 65.65 | 62.88 | 36.35 | 64.33 | 56.32 | 38.60 | 66.85 | 58.27 |
| LFT-Hassle-free-ALPS | | 15.81 | 11.57 | 18.44 | 74.59 | 64.91 | 64.06 | 36.52 | 64.56 | 54.51 | 37.80 | 68.69 | 58.20 |
| LFT-Hassle-free-ALPS-TM | | 15.37 | 11.22 | 17.90 | 74.70 | 66.45 | **64.56** | 38.48 | **66.22** | 55.60 | 38.60 | 66.09 | 58.84 |
| LFT-3BASiL | | 15.52 | 11.23 | 17.87 | 75.08 | 66.45 | 63.05 | 36.60 | 66.06 | 64.98 | **39.80** | 69.94 | **60.25** |
| LFT-3BASiL-TM | | **15.19** | **10.96** | **17.59** | 74.48 | **66.46** | 63.01 | **39.08** | 63.06 | 61.37 | 38.40 | 69.27 | 59.39 |
| **Llama-3.2-3B Dense** | – | 11.33 | 7.81 | 13.53 | 77.48 | 73.61 | 71.63 | 45.99 | 69.85 | 54.51 | 43.00 | 73.39 | 63.68 |

Table 11: (N:M Sparse + LR) decomposition performance for Llama-3.2-3B after LoRa Fine-Tuning (LFT). For Perplexity, (↓) lower values are better. For zero-shot tasks, (↑) higher values are better.

| Method | Config | C4 ↓ | WT2 ↓ | PTB ↓ | PIQA ↑ | HS ↑ | ARC-E ↑ | ARC-C ↑ | WG ↑ | RTE ↑ | OQA ↑ | BoolQ ↑ | Avg |
|---|---|---|---|---|---|---|---|---|---|---|---|---|---|
| LFT-OATS | | 37.46 | 31.49 | 49.71 | 62.24 | 36.88 | 38.72 | 24.23 | 51.93 | 52.71 | 27.20 | 62.02 | 44.49 |
| LFT-OATS-TM | | 28.23 | 23.86 | 36.25 | 65.61 | 43.81 | 42.09 | 25.26 | 52.64 | 52.71 | 30.60 | 64.13 | 47.11 |
| LFT-Hassle-free-SparseGPT | | 28.80 | 24.47 | 33.94 | 62.35 | 43.30 | 40.28 | 24.91 | 54.85 | 53.07 | 29.40 | 64.46 | 46.58 |
| LFT-Hassle-free-SparseGPT-TM | 2:8+64LR | 24.89 | 22.14 | 31.65 | 66.00 | 48.49 | 43.77 | 26.37 | 58.25 | 52.71 | 29.80 | 66.27 | 48.96 |
| LFT-Hassle-free-ALPS | | 25.31 | 21.97 | 31.99 | 66.38 | 48.42 | 43.73 | 26.62 | 59.43 | 53.43 | 29.60 | 47.16 | 46.85 |
| LFT-Hassle-free-ALPS-TM | | 22.85 | 20.23 | **28.00** | 67.95 | 52.18 | **46.84** | 27.99 | 60.06 | 53.43 | 31.20 | 61.44 | 50.20 |
| LFT-3BASiL | | 24.51 | 21.43 | 30.05 | 66.81 | 49.63 | 44.02 | 26.37 | 60.06 | **55.60** | 30.60 | 68.10 | 50.15 |
| LFT-3BASiL-TM | | **22.45** | **20.00** | 29.00 | **68.12** | **52.97** | 45.92 | 26.96 | **60.69** | 53.43 | **32.20** | **70.34** | **51.33** |
| LFT-OATS | | 17.87 | 12.65 | 20.59 | 72.36 | 61.20 | 57.45 | 35.58 | 63.93 | 53.07 | 35.00 | 70.58 | 56.15 |
| LFT-OATS-TM | | 16.18 | 11.43 | 18.34 | 73.61 | 65.43 | 60.65 | 37.46 | 66.54 | 57.40 | 36.80 | 75.72 | 59.20 |
| LFT-Hassle-free-SparseGPT | | 16.65 | 12.07 | 18.84 | 73.94 | 64.77 | 58.00 | 37.20 | 66.69 | 61.01 | 35.00 | 69.79 | 58.30 |
| LFT-Hassle-free-SparseGPT-TM | 3:8+64LR | 15.68 | 11.51 | 18.08 | 75.68 | 66.50 | 62.04 | 37.54 | 67.25 | **70.40** | 35.20 | 76.79 | **61.42** |
| LFT-Hassle-free-ALPS | | 15.92 | 11.77 | 17.89 | 74.59 | 66.86 | 60.35 | 36.43 | 68.67 | 65.34 | 37.20 | 74.71 | 60.52 |
| LFT-Hassle-free-ALPS-TM | | 15.29 | 11.46 | 17.74 | **75.90** | **68.34** | 61.66 | 37.03 | 69.33 | 59.57 | 36.20 | 69.33 | 59.99 |
| LFT-3BASiL | | 15.64 | 11.79 | 17.85 | 74.70 | 67.47 | **63.55** | 38.65 | 67.48 | 55.60 | **38.20** | 73.27 | 59.87 |
| LFT-3BASiL-TM | | **15.11** | **11.38** | **17.27** | 75.41 | 68.15 | 63.17 | **39.08** | 68.03 | 62.09 | 37.60 | **77.34** | 61.36 |
| LFT-OATS | | 13.09 | 8.67 | 14.64 | 77.80 | 72.31 | 70.92 | 45.39 | 70.72 | 60.65 | 40.40 | 75.60 | 64.22 |
| LFT-Hassle-free-SparseGPT | | 12.73 | 8.57 | 14.03 | 78.18 | 73.45 | 70.45 | 43.86 | 71.35 | 62.45 | 41.40 | 78.81 | 64.99 |
| LFT-Hassle-free-SparseGPT-TM | | 12.38 | 8.32 | 13.71 | 78.78 | 74.29 | **74.12** | 45.65 | 70.48 | 66.79 | 40.80 | 79.24 | 66.27 |
| LFT-Hassle-free-ALPS | 4:8+64LR | 12.55 | 9.44 | 14.45 | 76.82 | 71.19 | 71.04 | 44.45 | **72.77** | 56.68 | 40.20 | 78.13 | 63.91 |
| LFT-Hassle-free-ALPS-TM | | 12.31 | 8.29 | 13.43 | 78.84 | **74.92** | 73.86 | 48.29 | 71.19 | **68.59** | 40.80 | 79.51 | **67.00** |
| LFT-3BASiL | | 12.39 | 8.33 | 13.43 | 78.51 | 74.85 | 71.00 | 45.56 | 72.46 | 61.73 | **43.40** | 77.92 | 65.68 |
| LFT-3BASiL-TM | | **12.17** | **8.25** | 13.41 | **79.33** | 74.40 | 73.23 | **48.72** | 71.90 | 61.73 | 41.60 | 78.90 | 66.23 |
| LFT-OATS | | 14.34 | 9.67 | 16.24 | 77.15 | 69.54 | 65.66 | 40.70 | 68.27 | 66.43 | 38.80 | 73.73 | 62.54 |
| LFT-OATS-TM | | 13.46 | 9.09 | 14.81 | 77.69 | 71.25 | **69.99** | **44.28** | 69.77 | 58.84 | 40.60 | **78.07** | 63.81 |
| LFT-Hassle-free-SparseGPT | | 13.83 | 9.50 | 15.35 | 77.04 | 71.62 | 68.86 | 41.72 | 70.01 | 61.37 | 39.40 | 76.02 | 63.25 |
| LFT-Hassle-free-SparseGPT-TM | 2:4+64LR | 13.35 | 9.15 | 14.84 | **79.00** | 72.57 | 67.80 | 43.60 | 70.01 | 63.90 | 40.40 | 75.69 | 64.12 |
| LFT-Hassle-free-ALPS | | 13.49 | 9.29 | 14.61 | 77.48 | 72.40 | 67.09 | 42.92 | 70.56 | **67.15** | 39.80 | 77.13 | 64.32 |
| LFT-Hassle-free-ALPS-TM | | 13.20 | 9.12 | 14.34 | 78.24 | 72.51 | 67.38 | 41.89 | 70.80 | **67.15** | 40.40 | 75.63 | 63.67 |
| LFT-3BASiL | | 13.35 | 9.25 | 14.54 | 77.37 | 72.90 | 69.19 | 43.52 | 71.98 | **67.15** | **41.20** | 76.24 | **64.94** |
| LFT-3BASiL-TM | | **13.07** | **9.00** | **14.17** | 78.73 | **73.17** | 68.14 | 41.89 | 71.98 | 59.57 | 39.20 | 76.42 | 63.64 |
| **Meta-Llama-3-8B Dense** | – | 9.44 | 6.14 | 11.18 | 80.79 | 79.17 | 77.69 | 53.33 | 72.85 | 69.68 | 45.00 | 81.44 | 69.99 |

Table 12: (N:M Sparse + LR) decomposition performance for Meta-Llama-3-8B after LoRa Fine-Tuning (LFT). For Perplexity, (↓) lower values are better. For zero-shot tasks, (↑) higher values are better.

| Method | Config | C4↓ | WT2↓ | PTB↓ | PIQA↑ | HS↑ | ARC-E↑ | ARC-C↑ | WG↑ | RTE↑ | OQA↑ | BoolQ↑ | Avg |
|---|---|---|---|---|---|---|---|---|---|---|---|---|---|
| 3BASiL | | 25.31 | 20.59 | 28.33 | 71.27 | 55.56 | 54.25 | 30.80 | 64.56 | 53.43 | 33.20 | 72.91 | 54.50 |
| 3BASiL+OWL | | **23.21** | **19.54** | **27.09** | **71.71** | **57.90** | **56.69** | **33.53** | **67.56** | **54.15** | **33.40** | **77.86** | **56.60** |
| 3BASiL-TM | 70% + 64 | 19.55 | 16.44 | 21.64 | **74.16** | 59.76 | 58.04 | **33.28** | **66.38** | **58.12** | 35.20 | **73.15** | **57.26** |
| 3BASiL-TM+OWL | | **19.52** | **16.22** | **21.37** | 73.56 | **59.92** | **58.88** | 31.40 | 64.33 | 56.32 | **35.40** | 70.80 | 56.33 |
| 3BASiL | | 62.85 | 61.08 | 79.49 | 59.52 | 35.07 | 35.40 | 22.27 | 54.22 | 52.71 | 27.00 | 60.95 | 43.39 |
| 3BASiL+OWL | | **50.51** | **58.16** | **79.09** | **61.70** | **39.42** | **37.63** | **23.98** | **59.35** | **55.60** | **28.00** | **68.23** | **46.74** |
| 3BASiL-TM | 80%+64 | 36.51 | 39.32 | 57.94 | **65.07** | **42.05** | 39.94 | 25.00 | **58.80** | **52.71** | 26.00 | **64.59** | 46.77 |
| 3BASiL-TM+OWL | | **36.32** | **38.19** | **56.05** | 64.96 | 41.92 | **41.37** | **25.43** | 58.56 | **52.71** | **28.20** | 63.46 | **47.08** |
| **Meta-Llama-3-8B Dense** | – | 9.44 | 6.14 | 11.18 | 80.79 | 79.17 | 77.69 | 53.33 | 72.85 | 69.68 | 45.00 | 81.44 | 69.99 |

Table 13: Impact of OWL on `3BASiL` for (Unstructured + 64) decompositions of Meta-Llama-3-8B.

