# OpenReview forum: "3BASiL: An Algorithmic Framework for Sparse plus Low-Rank Compression of LLMs"
_NeurIPS.cc/2025/Conference — NeurIPS 2025 poster_

### Official Review · Reviewer_J8EX · 2025-06-25

**Clarity:** 3
**Significance:** 3
**Originality:** 3
**Rating:** 4
**Confidence:** 4

**Summary:**

In this paper, the authors propose 3BASiL, which introduces a unified algorithmic framework for compressing LLM by decomposing weight matrices into sparse + low rank form. Its key innovations include:

1) Propose a 3-block ADMM algorithm that jointly updates sparse, low-rank, and the dual variables with a closed-form update for each.

2) Propose a block-level weight fine-tune step that is able to further improve performance.

**Questions:**

Please see the weakness.

**Ethical Concerns:**

["NO or VERY MINOR ethics concerns only"]

**Final Justification:**

My concerns are addressed. I lean to accept.

**Limitations:**

The authors addressed the limitations in the Conclusion and limitation section.

**Paper Formatting Concerns:**

The equations between L117 and L122 should have indices.

**Quality:**

3

**Strengths And Weaknesses:**

**Strengths:**

1) The method offers closed-form updates for each component (sparse, low-rank, dual), and is supported by theoretical convergence guarantees.

2) Compared to prior work, it achieves up to 40% lower perplexity and 2–3× faster compression, demonstrating both efficiency and effectiveness.

**Weaknesses:**

1. The necessity of adding the term $\|\hat{W} - (S + L)\|$ in Equation (1) is not clearly explained. When computing updates, doesn't it behave similarly to $\|(X + \lambda)\hat{W} - (X + \lambda)(S + L)\|$?

2. While the method focuses on N:M sparsity, it seems generalizable. Have you evaluated its performance under unstructured or other structured sparsity formats? Additionally, could it adapt to non-uniform sparsity distributions, such as OWL[1]?

3. The paper mentions 80 alternating minimization steps during compression. Have you conducted sensitivity analysis on the number of steps?

[1] Lu et al. Outlier Weighed Layerwise Sparsity (OWL): A Missing Secret Sauce for Pruning LLMs to High Sparsity.

---

> ### Author Rebuttal · Authors · 2025-07-30
>
> We would like to thank the reviewer for the high-quality review, highlighting the strengths of our paper, and the useful suggestions to improve the manuscript that led to an insightful finding regarding using OWL for (S+LR) methods.
>
> **Weaknesses:**
>
> ***W1:*** You are making a very good point. If one expands the expression of equation (1), it is equivalent to minimizing $1/2Tr((\hat{W} - S - L)^T (X^TX+\lambda I) (\hat{W} - S - L))$. So it behaves like adding a regularization term to the local layer-wise reconstruction error Hessian $X^TX$. The aim here is to make the Hessian matrix $X^TX + \lambda I$ invertible (which is satisfied for any $\lambda > 0$) and reduce the numerical errors when applying equation (4).
>
> From a global optimization perspective, we aim to minimize the layer-wise reconstruction error $\textit{without deviating too far}$ from the original weights. It has also been used in previous (S + LR) compression methods like Hassle-free.
>
>
> ***W2.1:*** Our method is indeed generalizable to unstructured sparsity patterns. We experiment with a "less aggressive compression" configuration (0.5 + 128) and study the differences after further LoRA fine-tuning. We believe that it is interesting to further explore these settings as they are "near-lossless" configurations.
>
> Table 1: Comparison of the the perplexity of (S+LR) algorithms before and after LoRA fine-tuning on the configuration (0.5 + 128) on the model Llama3.2-1B. For perplexity (lower is better, ↓). If a dataset has the suffix "-LFT", we report the perplexity on that dataset after further LoRA fine-tuning on the C4 dataset [this can decrease the performance on WT2 and PTB datasets if the model has a performance comparable to dense model--3BASiL seems to be the only one to have this property].
>
>
> | **Method** | **Config** | **C4 ↓** | **C4-LFT ↓** | **WT2 ↓** | **WT2-LFT ↓** | **PTB ↓** | **PTB-LFT ↓** |
> |:---|:---:|:---:|:---:|:---:|:---:|:---:|:---:|
> | OATS | 0.5 + 128 | 17.99 | 16.71 | 12.29 | 11.76 | 21.68 | 20.97 |
> | Hassle-free-SparseGPT | 0.5 + 128 | 17.25 | 16.38 | 11.91 | 11.53 | 21.01 | 20.53 |
> | Hassle-free-ALPS | 0.5 + 128 | 16.81 | 16.12 | 11.62 | 11.41 | 20.59 | 20.54 |
> | 3BASiL | 0.5 + 128 | 16.17 | 15.69 | 11.17 | 11.13 | 19.84 | 20.00 |
> | 3BASiL-TM | 0.5 + 128 | **15.78** | **15.44** | **10.89** | **10.91** | **19.19** | **19.43** |
> | dense | -- | 14.01 | 14.01 | 9.75 | 9.75 | 17.59 | 17.59 |
>
>
> ***W2.2:*** Our method can also support non-uniform sparsity distributions and we have added support for OWL for unstructured sparsity. This is a great insight and we would like to thank the reviewer for raising this important remark. We have tried to activate OWL for the configuration (0.7 + 64) for Llama3-8B. It turns out that this does improve the results of 3BASiL. This is not trivially true because OWL deltas were optimized for pure pruning methods. We believe that this is an insightful finding for the community. It opens further research directions as to how to create similar deltas for the low-rank components, so thanks to the reviewer for this suggestion!
>
> Table 2: Comparison of the performance of our proposed method 3BASiL with and without activating OWL on the configuration (0.7 + 64) on Llama3-8B. For perplexity (lower is better, ↓). For accuracy (higher is better, ↑).
>
> | **Method** | **Config** | **C4 ↓** | **WT2 ↓** | **PTB ↓** | **PIQA ↑** | **ARC-E ↑** | **ARC-C ↑** | **BoolQ ↑** | **HellaSwag ↑** | **OpenBookQA ↑** | **RTE ↑** | **WinoGrande ↑** | **Avg ZS ↑** |
> |:---|:---:|:---:|:---:|:---:|:---:|:---:|:---:|:---:|:---:|:---:|:---:|:---:|:---:|
> | 3BASiL | 0.7 + 64 | 25.33 | 20.51 | 28.01 | 71.16 | 53.49 | 31.83 | 74.22 | 55.32 | 32.60 | **55.96** | 65.90 | 55.06 |
> | 3BASiL-OWL | 0.7 + 64 | **23.32** | **19.84** | **27.09** | **71.93** | **59.81** | **32.68** | **74.71** | **58.50** | **33.80** | 53.79 | **66.22** | **56.43** |
>
> We intend to add a section showcasing the value of activating OWL for highly sparse unstructured configuration [e.g. (0.7 + 64)] and adding a discussion motivating the question of non-uniform sparsity and non-constant rank allocation as a future research direction.
>
> ***W3:*** 80 alternating minimization steps is the default value in the papers of OATS and Hassle-free. A sensitivity analysis has been conducted in OATS for instance [see figure 1 of OATS paper [1]], and shows that the performance plateaus after 80 minimization steps. For these alternating-minimization based methods, increasing the number of steps generally lowers the objective (1) and improves the performance. If we were to reduce the number of steps, the performance would degrade.If we were to increase it, the performance gain is negligible (plateau reached) in our experiments but the runtime still increases.
>
> Our algorithm is different from alternating minimization [used by OATS/HASSLE-free] and the number of iterations is not directly comparable to OATS and HASSLE-free. That being said, 3BASiL is both faster and achieves a better performance than OATS and HASSLE-free [see figure 2 in our submission].
>
>
> [1] Zhang et al., OATS: Outlier-Aware Pruning Through Sparse and Low Rank Decomposition (ICLR'25)

---

> > ### Comment · Reviewer_J8EX · 2025-08-04
> > **Official comment by Reviewer J8EX**
> >
> > Thank you for the authors’ response.
> >
> > Most of my concerns have been adequately addressed. I will maintain my score, which leans toward acceptance.

---

> > > ### Author Response · Authors · 2025-08-05
> > >
> > > We sincerely appreciate your time and valuable feedback, which helped us improve the paper. Thank you for maintaining your positive assessment of our work!

---

### Official Review · Reviewer_UVpP · 2025-06-27

**Clarity:** 3
**Significance:** 3
**Originality:** 3
**Rating:** 5
**Confidence:** 4

**Summary:**

This paper proposes a sparse + low-rank approach for model compression and introduces cross-layer joint optimization to improve the performance of the compressed model.

**Questions:**

1.The paper primarily presents experimental results for the semi-structured sparse component. To demonstrate the method's general applicability, could additional comparisons with unstructured sparsity be attempted?

2.Could the actual acceleration effects of the compressed model in practical usage be provided to show that the low-rank + sparse method is indeed well-adapted to hardware?

3.If conditions permit, a more extensive assessment can be conducted appropriately.

**Ethical Concerns:**

["NO or VERY MINOR ethics concerns only"]

**Final Justification:**

These responses have addressed most of my concerns. Given that the current evaluation is already positive, we will not modify the score.

**Limitations:**

Yes

**Quality:**

3

**Strengths And Weaknesses:**

**Strengths**：

1.This work proposes a novel three-block variable decomposition framework that jointly optimizes sparsity and low-rank constraints, which address the theoretical limitations of existing alternating minimization methods.

2.Introduced a TM fine-tuning step after model compression to enhance the performance of the compressed model.

**Weaknesses**：

1.This method is only applicable to semi-structured conditions, which limits its scope of application

2.Experiments were only conducted on models of 1-8B size. The performance on larger-scale models remains to be further verified

---

> ### Author Rebuttal · Authors · 2025-07-30
>
> We would like to thank the reviewer for the extensive feedback, highlighting the strengths of our paper, and the valuable suggestions to improve the manuscript.
>
>
> **Weaknesses and Questions**
>
> ***W1 & Q1:*** Our method is indeed generalizable to unstructured sparsity patterns. We experiment with a "less aggressive compression" configuration (0.5 + 128) and study the differences after further LoRA fine-tuning. We believe that it is interesting to further explore these settings as they are "near-lossless" configurations.
>
> Table 1: Comparison of the the perplexity of (S+LR) algorithms before and after LoRA fine-tuning on the configuration (0.5 + 128) on the model Llama3.2-1B. For perplexity (lower is better, ↓). If a dataset has the suffix "-LFT", we report the perplexity on that dataset after further LoRA fine-tuning on the C4 dataset [this can decrease the performance on WT2 and PTB datasets if the model has a performance comparable to dense model--3BASiL seems to be the only one to have this property].
>
> | **Method** | **Config** | **C4 ↓** | **C4-LFT ↓** | **WT2 ↓** | **WT2-LFT ↓** | **PTB ↓** | **PTB-LFT ↓** |
> |:---|:---:|:---:|:---:|:---:|:---:|:---:|:---:|
> | OATS | 0.5 + 128 | 17.99 | 16.71 | 12.29 | 11.76 | 21.68 | 20.97 |
> | Hassle-free-SparseGPT | 0.5 + 128 | 17.25 | 16.38 | 11.91 | 11.53 | 21.01 | 20.53 |
> | Hassle-free-ALPS | 0.5 + 128 | 16.81 | 16.12 | 11.62 | 11.41 | 20.59 | 20.54 |
> | 3BASiL | 0.5 + 128 | 16.17 | 15.69 | 11.17 | 11.13 | 19.84 | 20.00 |
> | 3BASiL-TM | 0.5 + 128 | **15.78** | **15.44** | **10.89** | **10.91** | **19.19** | **19.43** |
> | dense | -- | 14.01 | 14.01 | 9.75 | 9.75 | 17.59 | 17.59 |
>
>
> ***W2 & Q2:*** Thank you for pointing out this very important point! We are currently running experiments on the model OPT-30B which we intend to include in the revised paper [see Table 2 below for a comparison between 3BASiL and Hf-SparseGPT--it seems that this type of compression has less impact on performance for larger models].
>
> According to SLOPE [1], we should expect an inference acceleration of up to 1.25× and up to 0.63 memory reduction for this model if we use their custom designed CUDA kernels during inference.
>
> Table 2: Comparison of the the perplexity of Hf-SparseGPT and 3BASiL on (2:4 + 64) configuration for a OPT-30B model.
>
> | **Method** | **Config** | **C4 ↓** | **WT2 ↓** | **PTB ↓** |
> |:---|:---:|:---:|:---:|:---:|
> | Hf-SparseGPT | 2:4 + 64 | 11.64 | 10.34 | 14.54 |
> | 3BASiL | 2:4 + 64 | **11.55** | **10.04** | **14.32** |
> | Dense | - | 11.44 | 9.56 | 14.04 |
>
>
> ***Q3:*** In the revised paper, we are going to expand our experiments to report more unstructured sparsity (S+LR) configurations similar to Table 1 above. We intend to include more pure pruning results to showcase the universality of Transformer Matching (see Table 2 of our response to reviewer JUTM). In addition, we are going to include a subsection showcasing the applicability of our method to non-uniform transformer compression (see Table 2 of our response to reviewer J8EX). Finally, we will report some results on the model OPT-30B to see the performance of 3BASiL on larger models [similar to Table 2 above].
>
> [1] Mozaffari et al., SLoPe: Double-Pruned Sparse Plus Lazy Low-Rank Adapter Pretraining of LLMs (ICLR'25).

---

> > ### Comment · Reviewer_UVpP · 2025-08-04
> >
> > These responses have addressed most of my concerns. Given that the current evaluation is already positive, we will not modify the score.

---

> > > ### Author Response · Authors · 2025-08-05
> > >
> > > We sincerely appreciate your time and valuable feedback, which helped us improve the paper. Thank you for maintaining your positive assessment of our work!

---

### Official Review · Reviewer_pjgM · 2025-06-28

**Clarity:** 3
**Significance:** 3
**Originality:** 2
**Rating:** 5
**Confidence:** 4

**Summary:**

In this paper, an ADMM algorithm is studied for compressing dense weights into sparse-plus-low-rank (S+LR) representations. With particular sparse patterns, such as 2:4 sparsity, S+LR matrices can be computed efficiently on GPUs. The paper proposes a multi-block ADMM procedure to initialize the factors using calibration data, which provides a good initialization and thus improves the model quality after fine-tuning. Each factor is obtained by minimizing the augmented Lagrangian while fixing the others, with convergence guarantees as the penalty term rho increases. To address misalignment in layer-wise output matching, the paper also introduces module-wise output matching. Experimental results demonstrate that 3BASiL improves both one-shot and fine-tuned compression outcomes across various structured sparsity patterns on large language models.

**Questions:**

1. Why is 3BASiL designed with a 3-block ADMM? In the Robust PCA literature, an ADMM algorithm to solve robust PCA $\min_{L,S} ||L||_* + ||S||_1$  s.t.  $L+S=W$ can be formulated as a 2-block method. Is the use of 3 blocks in 3BASiL due to the hard sparsity constraint, unlike the l1 loss in RPCA?
2. The last inequality of Equation 18 uses $||S^{(t+1)} - S^{(t)}||_F \le \frac{C_A}{\rho_t}$, whereas the bound in Equation 17 is $\frac{3C_A}{\rho{t-1}}$. How is the former inequality obtained? Can’t the bound in Equation 17 be used directly?
3. Transformer-level Matching is proposed to update the factors with a better proxy than layer-level matching. If the quality of this proxy is a concern, couldn’t the S and L factors be updated directly based on the original training loss (e.g., cross-entropy) or a knowledge distillation loss on a subset of C4, instead of splitting the process into Transformer-level Matching and LoRA fine-tuning?

**Ethical Concerns:**

["NO or VERY MINOR ethics concerns only"]

**Final Justification:**

All of my concerns are resolved.

**Limitations:**

Yes

**Quality:**

3

**Strengths And Weaknesses:**

**Strengths**

1. The paper presents both theoretical and practical contributions. The proposed algorithm appears to outperform other S+LR decomposition models, and the experimental results show that the compressed models maintain decent performance. Theorem 1 provides convergence guarantees, which is a strong point.
2. The writing quality is good, making the paper easy to understand.

**Weaknesses**

Most of my concerns relate to the experimental results.

1. Although Theorem 1 proves convergence of the proposed ADMM steps, it is unclear whether convergence occurs in practice. Please consider including plots showing the diminishing difference between L and S at each step or the convergence of the augmented Lagrangian.
2. The actual inference speedup is not reported. While it seems likely that the compressed network will run efficiently on GPUs supporting 2:4 or 4:8 sparsity, reporting real inference speed measurements would help emphasize the practical impact of the proposed method.

---

> ### Author Rebuttal · Authors · 2025-07-31
>
> We would like to thank the reviewer for the thorough review, for highlighting the strengths of our submission, and for raising important fundamental questions about our compression algorithm.
>
> **Weaknesses & Questions**
>
> ***W1:*** Thanks for pointing out this remark. We also think that including plots for the convergence of the objective and iterates will further improve the quality of the paper, we will add such plots in the revised version.
>
> Since we are not allowed to include plots for this rebuttal per NeurIPS policy, we wanted to share simple logs we used to follow the convergence of 3BASiL [for the compression of self_attn.q_proj of the first transformer of Llama3-8B under a (2:4 + 128) configuration].
>
> For iteration 009, the true loss ||X(W_old - (W_S + B @ A))||_F^2 = 1952.1258544921875
>
> For iteration 049, the true loss ||X(W_old - (W_S + B @ A))||_F^2 = 664.7432861328125
>
> For iteration 149, the true loss ||X(W_old - (W_S + B @ A))||_F^2 = 176.23971557617188
>
> For iteration 199, the true loss ||X(W_old - (W_S + B @ A))||_F^2 = 119.56678771972656
>
> For iteration 299, the true loss ||X(W_old - (W_S + B @ A))||_F^2 = 87.93989562988281
>
> For iteration 399, the true loss ||X(W_old - (W_S + B @ A))||_F^2 = 87.01194763183594
>
> For iteration 479, the true loss ||X(W_old - (W_S + B @ A))||_F^2 = 86.82907104492188
>
> [479 is the last iteration, note that iterations in our algorithm are extremely efficient and are not comparable to the default value 80 of alternating minimization steps used in HASSLE-free and OATS].
>
> ***W2:*** Thank you for pointing out this very important point! We are currently running experiments on the model OPT-30B which we intend to include in the revised paper [see Table 2 below for a comparison between 3BASiL and Hf-SparseGPT--it seems that this type of compression has less impact on performance for larger models].
>
> According to SLOPE [1], we should expect an inference acceleration of up to 1.25× and up to 0.63 memory reduction for this model if we use their custom designed CUDA kernels during inference.
>
> Table 2: Comparison of the the perplexity of Hf-SparseGPT and 3BASiL on (2:4 + 64) configuration for a OPT-30B model.
>
> | **Method** | **Config** | **C4 ↓** | **WT2 ↓** | **PTB ↓** |
> |:---|:---:|:---:|:---:|:---:|
> | Hf-SparseGPT | 2:4 + 64 | 11.64 | 10.34 | 14.54 |
> | 3BASiL | 2:4 + 64 | **11.55** | **10.04** | **14.32** |
> | Dense | - | 11.44 | 9.56 | 14.04 |
>
> ***Q1:*** Thank you for making the connection between our optimization formulation and the RPCA literature. Our problem is quite different from RPCA as we aim to match the $\textbf{outputs}$ of each weight matrix $||X(W - S - L)||_F$ as opposed to the actual matrix weights matching $||W - S - L||_F$. A block in ADMM is introduced to this end. Please also note [line 154 of our paper] that our 3-block ADMM approach can be reformulated as a standard 2-block because the Lagrangian is separable with respect to 2 blocks introduced for the optimization.
>
> ***Q2:*** This is a typo, thanks for catching it. The correct inequality should be
>
> $| L^{(t+1)} - L^{(t)} |_F \leq \frac{3C_F^2 C_A}{\rho_{t-1}}$,
>
> which directly uses the bound in Eq (17). The constant $C$ should be correspondingly changed to
> $C=3C_F^2C_A$
> (since $C_F\ge 1$). We deeply appreciate the reviewer's careful examination and have corrected this typo in our revised manuscript.
>
> ***Q3:*** This is a fundamental question related to one-shot compression methods! It is true that $S$ and $L$ can be updated with the original loss function ($L$ is updated during LoRA fine-tuning for example) or a knowledge distillation loss from the original model. However, if one aims to update $S$, one needs to do a full back-propagation on the entire LLM which is very memory intensive. Our entire pipeline: compression of LLama3-8B with 3BASiL, refinement with TM and LoRA fine-tuning can be done on a single A100 GPU. However, full back-propagation even with batch=1 results in a cuda memory error. That's why Transformer Matching introduces an intermediate [memory-efficient] loss function which provides is a good trade-off between the local-layerwise reconstruction loss and the original training loss.
>
>
> [1] Mozaffari et al., SLoPe: Double-Pruned Sparse Plus Lazy Low-Rank Adapter Pretraining of LLMs (ICLR'25).

---

> > ### Comment · Reviewer_pjgM · 2025-08-04
> >
> > I appreciate your thorough response.
> >
> > The loss log makes sense and highlights that the algorithm works in practice. Thank you for providing it!
> >
> > Regarding the inference speedup, does the 2:4+r64 3BASiL configuration correspond to the row "OPT-30B" and the column "1.56% Adapter"? It seems like the expected speedup is 1.53x in Table 1 of [1]. Could the author provide a pointer to the reference value in [1]?
> >
> > I think this is a strong paper. I'll keep my score the same as I have already championed the paper towards acceptance.
> >
> > [1] Mozaffari et al., SLoPe: Double-Pruned Sparse Plus Lazy Low-Rank Adapter Pretraining of LLMs (ICLR'25).

---

> > > ### Author Response · Authors · 2025-08-05
> > >
> > > The rank 64 corresponds to ~0.9% (hidden dimension is 7168 for OPT-30B) and hence we expect improved speedups and memory savings compared to the reference value of 1.56% Adapter.
> > > We will launch the same experiments with the rank 112 corresponding to exactly 1.56% and report the reference value in [1] in the revised manuscript.
> > >
> > > We sincerely appreciate the time you took to review our work and the positive feedback you had about our submission, we believe that it has improved the quality of the submission!
> > >
> > > [1] Mozaffari et al., SLoPe: Double-Pruned Sparse Plus Lazy Low-Rank Adapter Pretraining of LLMs (ICLR'25).

---

### Official Review · Reviewer_JUTM · 2025-07-03

**Clarity:** 4
**Significance:** 4
**Originality:** 4
**Rating:** 6
**Confidence:** 4

**Summary:**

The large scale of Large Language Models (LLMs) makes them a big target for compression, and one common technique is sparsity which, when coupled with low-rank decomposition, shows promise at matching the dense model's quality.  Despite many developments in this direction, there still exists a large gap in model quality.  The authors present 3BASiL-TM to help close this gap.  First, 3BASiL is a 3-block ADMM algorithm which alternates between updating the sparse version of the weights, the low-rank component, and the sparse mask.  Treating L and D as a single variable block reformulates the approach as standard 2-block ADMM and helps guarantee convergence.  Next, the authors propose Transformer-level Matching (denoted by a -TM suffix), which serves to further refine the sparse and low-rank components, using the nonlinear behavior of each transformer block, so that the low-rank components are well-initialized for LoRA adaptation.  Empirical studies show that TM generalizes to many one-shot S+LR methods, that its application to 3BASiL delivers top-quality results, and that the benefit of 3BASiL-TM is maintained, even after LoRA fine-tuning.

**Questions:**

These two questions directly relate to the quality of the submission:
1. How does TM improve, say, the 2:4 and 4:8 results in Table 3?
2. How does EoRA interact with 3BASiL as a whole, and specifically with using TM as a smart initialization for LoRA fine-tuning?

**Ethical Concerns:**

["NO or VERY MINOR ethics concerns only"]

**Final Justification:**

The authors' rebuttal entirely satisfactorily answered my questions, and reading other reviews and responses has not changed my initial review's conclusions.

**Limitations:**

The limitations are satisfactorily discussed in Section 6.

**Paper Formatting Concerns:**

No concerns

**Quality:**

3

**Strengths And Weaknesses:**

## Strengths
This submission is outstanding.  It presents a novel collection of related ideas, clearly motivating and proving the benefit of each.  Treating each metric in turn:
### Quality
I couldn't find any missing links in the chain of reasoning from claim to claim.  Each claim was supported by theory or empirical results, as appropriate, with one exception (noted below).  This is clearly a complete package and not a work in progress.  (Future work is indicated, but well beyond the scope of a single endeavor.)
### Clarity
I believe that sufficient detail is presented such that a dedicated practitioner could implement the technique and reproduce results.  Fully understanding each step does take careful reading, but the organization and exposition is not to blame; the mechanisms used are simply nontrivial.
### Significance
3BASiL-TM represents a major step forward in recovering model quality.  There are still gaps to the dense baselines, but they are greatly reduced without undue computational load. Care has been taken to not only find compelling new techniques, but to make them practical for use - runtimes in single-digit hours is entirely reasonable that users shouldn't need to shy away from one-time applications to their models due to limited resources.  (Larger models will require more time, but will also require more resources to train and deploy.)  As I discuss immediately below, there are findings within the results that I consider very useful for advancing the field.
### Originality
I only noted one missing piece of related work (below); otherwise, the comparison to prior work is complete and shows that the collection of techniques in 3BASiL-TM is both new and important.

A couple observations show the importance of the submission goes beyond its ability to improve zero-shot model quality:
- Table 3 is particularly interesting: for a given compression rate, it's important to find the right balance between sparsity and the rank of the LR.
- Figure 4 is also important: I've observed "head-starts" by clever one-shot schemes become washed out after fine-tuning.  It's good to see this doesn't happen to 3BASiL-TM (at least for the limited fine-tuning performed).

## Weaknesses
I noted one missing comparison - EoRA (1), which seems like it might compete with the "smart initialization" of LoRA parameters offered by transformer matching.

I also could not find evidence to support the claim on line 181 that TM can apply to sparse, but not LR, networks.  I can imagine how it will apply, but this is not supported in any of the presented results.

Minor issues:
- Inconsistent citation styles are confusing - particularly (but not limited to) lines 213-221.  Some citations are missing years (e.g. lines 290, 292).
- The wrong entry in Table 2's 2:4+64LR RTE results is bolded (or there's a typo in one of the 3BASiL results).
- Citing the N:M sparse format (and the hardware advancements that make it useful) (e.g., 2) would make the background information complete.

1. Liu et al., EoRA: Fine-tuning-free Compensation for Compressed LLM with Eigenspace Low-Rank Approximation, https://arxiv.org/abs/2410.21271
2. Mishra et al., Accelerating Sparse Deep Neural Networks, https://arxiv.org/abs/2104.08378

---

> ### Author Rebuttal · Authors · 2025-07-30
>
> We would first like to thank the reviewer for the thorough examination of the paper, highlighting the strengths of our submission and proposing missing pieces that would improve the manuscript.
>
> **Weaknesses -- Questions**
>
> ***W1 & Q2:*** Thank you very much for pointing EoRA which should indeed be included in the related works of our paper. Upon inspection of this important method, we noted that HASSLE-free [a considered competing method] with a special configuration (alternating minimization steps=1 and using our improved implementation--see Table 4 discussion) reduces to EoRA. In that case, Hf-SparseGPT-ours would (i) compress the model weights to 2:4 sparsity using the SparseGPT algorithm and (ii) use the closed-form rank-update (equation 4) for "compensation" as discussed in the EoRA paper [1]. In EoRA, $Q^\prime$ corresponds to $H^{1/2}$ in our paper (up to a minor notation difference, as they consider $XX^T$ to be the hessian whereas we use $X^TX$).
>
> We launched experiments with EoRA, Hf-SparseGPT-ours (default AM steps=80) and 3BASiL for Llama3-8B under a (2:4 + 128) configuration [similar to Table 2 in EoRA table] (we also compare the effect of using TM on top of EoRA to show universality).
>
> We obtain the following results:
>
> Table 1: Comparison of (S+LR) methods with (2:4 + 128) configuration on Llama3-8B. For perplexity (lower is better, ↓). For accuracy (higher is better, ↑).
>
> | Method              | C4 PPL ↓ | WT2 PPL ↓ | PTB PPL ↓ | ARC-C ↑ | HellaSwag ↑ | WinoGrande ↑ | Avg. ↑ |
> |:--------------------|:--------:|:---------:|:---------:|:-------:|:-----------:|:------------:|:------:|
> | EoRA                | 20.34    | 14.49     | 21.86     | 35.75   | 61.02       | 67.17        | 58.23  |
> | EoRA-TM             | 16.00        | 11.41         | 17.32         | 38.82   | 65.11       | 66.54        | 59.91  |
> | Hf-SparseGPT-ours   | 15.46        |10.50         | 16.05         | 40.61   | 69.27       | 70.72        | 63.30  |
> | 3BASiL              | 14.45    | 9.56     | 14.46     |43.94   | 71.29       | 70.65        | 65.06  |
> | **3BASiL-TM** | **13.30**| **8.82** | **13.92** | **44.88**| **72.28** | **71.59** | **65.40** |
>
> Note that the results of our implementation of EoRA are slightly different than reported in the EoRA paper.
>
> (i) Our EoRA implementation has a slightly worse WikiText ppl than reported in the EoRA paper (11.07) because our calibration data (X matrix) is C4 whereas they consider WikiText2 (training data). Despite this fact, 3BASiL-TM [with calibration C4] largely outperforms EoRA [with calibration WikiText2] on the WikiText2 test perplexity task (from 11.07 to 8.82) under the same constraints.
>
> (ii) Our EoRA implementation has a slightly better ARC-C accuracy than reported in the original EoRA paper. For this metric, we both use the same calibration data C4. That being said, we use more calibration data (128 segments vs. 64 in EoRA paper). Our method 3BASiL-TM improves the accuracy post-compression from 35.75 to 44.88.
>
>
> We would like to thank the reviewer again for pointing us to this important method. We intend to add it to the related works discussion (Exact Low-Rank updates for layer-wise compression). We also think it is helpful for the compression community if we add a paragraph discussing the underlying connections between our improved implementation of Hf-SparseGPT-ours and EoRA as well as more experiments in the Appendix.
>
> ***W2 & Q1:***
> This is a very good remark. We have launched preliminary results for the universality of Transformer Matching to pure pruning methods. The results in Table 3 of the paper for pure pruning methods have been extracted from HASSLE-free paper [2]. During the rebuttal, we have added support for SparseGPT and ALPS with TM. The results are below.
>
> Table 2: Comparison of pure pruning methods with 2:4 and 4:8 configurations on Llama3-8B. For perplexity (lower is better, ↓). For accuracy (higher is better, ↑).
>
> | **Method** | **Config** | **C4 ↓** | **WT2 ↓** | **PTB ↓** | **PIQA ↑** | **ARC-E ↑** | **ARC-C ↑** |
> |:---|:---:|:---:|:---:|:---:|:---:|:---:|:---:|
> | SparseGPT | 2:4 | 22.62 | 16.14 | 25.31 | 71.49 | 56.19 | 34.04 |
> | SparseGPT-TM | 2:4 | 15.31 | 10.82 | 17.07 | 76.50 | 66.04 | 40.70 |
> | ALPS | 2:4 | 19.80 | 14.59 | 22.42 | 73.56 | 59.60 | 35.15 |
> | ALPS-TM | 2:4 | **14.99** | **10.69** | **16.53** | **77.31** | **66.88** | **40.87** |
> |---|---|---|---|---|---|---|---|
> | SparseGPT | 4:8 | 17.56 | 12.27 | 18.63 | 74.86 | 62.92 | 38.82 |
> | SparseGPT-TM | 4:8 | 13.68 | 9.25 | 14.51 | 77.97 | **69.99** | **44.11** |
> | ALPS | 4:8 | 16.08 | 11.23 | 16.58 | 75.90 | 65.07 | 40.27 |
> | ALPS-TM | 4:8 | **13.61** | **9.18** | **14.28** | **78.35** | 69.74 | 42.75 |
>
>
> We agree with the reviewer that further experiments should be included to pure sparsity algorithms to showcase the universality of our proposed Transformer Matching procedure. In the revised paper, we will add support to all reported pruning methods and their enhanced TM versions.
>
>
> ***W3:***
> Thank you for pointing out these issues. They have been fixed in the revised manuscript.
>
> We want to reiterate our appreciation for the thorough feedback of the reviewer. We believe that incorporating the suggestions and addressing the questions and weaknesses discussed above will greatly improve the quality of the revised paper.
>
>
>
> [1] Liu et al., EoRA: Fine-tuning-free Compensation for Compressed LLM with Eigenspace Low-Rank Approximation.
>
> [2] Makni et al., A unified framework for Sparse plus Low-Rank Matrix Decomposition for LLMs (CPAL'25).

---

> > ### Comment · Reviewer_JUTM · 2025-08-04
> >
> > I thank the authors for their response, which I find adds to the already very strong submission.  I'll keep my score at a strong accept.

---

> > > ### Author Response · Authors · 2025-08-05
> > >
> > > We sincerely appreciate the time you took to review our work and are glad that our revisions addressed your concerns!

---

### Note · Authors · 2025-08-14

We sincerely thank the reviewers and area chairs for their time, effort, and insightful feedback. We are encouraged by the positive assessments received, noting in particular the significance of the performance gains and runtime improvements of our 3-block ADMM approach (JUTM, J8EX), the theoretical convergence guarantees (all reviewers), and appreciation of our memory-efficient universal transformer-matching TM (JUTM, UVpP).

****Rebuttal Summary****

We are delighted that our responses and new experiments were well-received and addressed the reviewers' concerns. These are the main points discussed during the rebuttal period.

****Universality of transformer-matching to pure-pruning (JUTM).**** Reviewer JUTM directed us to add experiments showcasing the universality of our memory-efficient transformer-matching step to pure pruning method, we have added support for SparseGPT and ALPS. We see large improvement gains thanks to TM. More pure pruning results have been added to the revised manuscript.

****Actual model speedups post compression (pjgM) & Larger model integration (UVpP).**** To strengthen the submission, reviewer pjgM suggested measuring the real inference speed post-compression and reviewer UVpP suggested scaling our compression to even larger LLMs (than initial 8B models). We provided results for the compression of OPT-30B models into a (2:4 + 64) configuration with minimal performance degradation. This shows the scaling of our method to 30B models. The runtime gains for such compression should be at least x1.53, a reference value reported in a paper from related works.

****Unstructured sparsity (UVpP, J8EX).**** During the rebuttal, we provided experiments for the configuration (0.5 + 128) for Llama3.2-1B. We have also added results under the same configuration for all considered Llama models in the revised manuscript.

****OWL integration (J8EX).**** Reviewer J8EX noted that our method is flexible and could support non-uniform sparsity distributions across transformers. During the rebuttal, we have added support for OWL for Llama3-8B and it does improve the performance over uniform sparsity allocation. This serves as a strong baseline for future research aiming to allocate different sparsity/rank patterns for (S+LR) methods.

We thank all reviewers again for their thorough assessment, pointing out typos, and suggesting ideas which we think have strengthened the revised paper a lot.

---

### Decision · Program_Chairs · 2025-09-17

**Decision:**

Accept (poster)

**Comment:**

The paper proposes 3BASiL, an efficient one-shot post-training method for decomposing large language models (LLMs). Section 2 formulates the optimization problem, which is solved using the ADMM framework. A key component of the method is the Transformer Matching (TM) step, which jointly optimizes all sparse and low-rank components across layers within a transformer block to better approximate the original model’s output. The paper presents an extensive set of experiments demonstrating the effectiveness of the proposed approach.

All reviewers appreciated the novelty of the method and the rigor of the experimental evaluation. Initial concerns raised during the review process were addressed during the discussion phase, including clarifications and additional results. These responses were well-received, and there was broad consensus among reviewers that the paper meets the standards for acceptance.

I recommend acceptance. The paper offers a well-motivated and technically sound contribution, supported by thorough experimentation. For the camera-ready version, the authors should incorporate all requested changes, including the additional experiments presented during the rebuttal.